# *Let's Roll a BiFTA*: Bi-refinement for Fine-grained Text-visual Alignment in Vision-Language Models

**Yuhao Sun**                                                    *yuhao.sun1@student.unimelb.edu.au*
*School of Computing and Information Systems*
*The University of Melbourne*

**Chengyi Cai**                                                          *chengyi.cai1@unimelb.edu.au*
*School of Computing and Information Systems*
*The University of Melbourne*

**Jiacheng Zhang**                                                  *jiacheng.zhang6@unimelb.edu.au*
*School of Computing and Information Systems*
*The University of Melbourne*

**Zesheng Ye**                                                            *zesheng.ye@unimelb.edu.au*
*School of Computing and Information Systems*
*The University of Melbourne*

**Xingliang Yuan**                                                    *xingliang.yuan@unimelb.edu.au*
*School of Computing and Information Systems*
*The University of Melbourne*

**Feng Liu**[*]                                                            *fengliu.ml@gmail.com*
*School of Computing and Information Systems*
*The University of Melbourne*

**Reviewed on OpenReview:** *https://openreview.net/forum?id=ZmbkzZnHO4*

## Abstract

Recent research has shown that aligning fine-grained text descriptions with localized image patches can significantly improve the zero-shot performance of pre-trained vision-language models (e.g., CLIP). However, we find that both fine-grained text descriptions and localized image patches often contain redundant information, making text-visual alignment less effective. In this paper, we tackle this issue from two perspectives: *View Refinement* and *Description refinement*, termed as **Bi**-*refinement for* **F**ine-grained **T**ext-visual **A**lignment (BiFTA). *View refinement* removes redundant image patches with high *Intersection over Union* (IoU) ratios, resulting in more distinctive visual samples. *Description refinement* removes redundant text descriptions with high pairwise cosine similarity, ensuring greater diversity in the remaining descriptions. BiFTA achieves superior zero-shot performance on 6 benchmark datasets for both ViT-based and ResNet-based CLIP, justifying the necessity to remove redundant information in visual-text alignment.

## 1 Introduction

Drawing from the profound strides made in large-scale pre-training within natural language processing (Radford et al., 2018; 2019; Devlin et al., 2019; Brown et al., 2020), the CLIP model (Radford et al., 2021)

---

[*]Corresponding author.
  Code is available at `https://github.com/tmlr-group/BiFTA`.

aligns vast collections of images with their corresponding natural language captions (e.g., "a photo of a {label}") into a unified embedding space using large datasets. The scaling of the pre-training data in CLIP empowers the model to deliver considerable performance in zero-shot classification (Radford et al., 2021). CLIP performs zero-shot classification by computing cosine similarity scores between image representations and textual label prompts so that the final prediction is determined based on the distribution of these similarity scores (He & Peng, 2017; Liang et al., 2020; Radford et al., 2021; Yan et al., 2025; Mahajan et al., 2025). To push the limits of CLIP's zero-shot capabilities, several studies (Menon & Vondrick, 2023; Pratt et al., 2023) leverage the generative capabilities of *Large Language Models (LLMs)* to produce fine-grained, label-specific textual descriptions via prompt templates (e.g., "describe what does a/an {label} look like"). Under this framework, CLIP model computes and integrates the cosine similarity scores between visual representation of the input image and each generated textual descriptions. By enriching the textual modality, these refined prompts significantly boost CLIP's zero-shot classification accuracy. Recently, Li et al. (2024) propose ***w**eighted visual-text **c**ross **a**lignment score* (WCA), which calculates and integrates a weighted cross-alignment score between LLM-generated label-specific descriptions and cropped image patches. By effectively refining the synergy between visual and textual representations, this **cross-alignment scoring** mechanism achieves the *state-of-the-art* (SOTA) zero-shot performance on several downstream tasks.

However, as demonstrated in Figure 1, we observe that both the LLM-generated textual descriptions and the randomly cropped image patches often exhibit redundant content. Such redundancy may introduce disproportionate contributions to the cross-alignment score computation, thereby compromising the accuracy of the resulting alignment. In specific, following WCA (Li et al., 2024), we use random cropping to obtain localized image patches of an image sample (e.g., a border collie), as depicted in Figure 1. We find that these image patches often include certain redundant views exhibiting exceptionally high pairwise cosine similarities (i.e., approaching to 1), which is attributed to the randomness in the cropping method. Similarly, we observe that the diversity of LLM-generated textual descriptions is often restricted by the invariant label-integrated prompt template, which directly leads to generate redundant text responses (see Figure 1). In Figure 2, we further examine how the aforementioned redundancies affect the accuracy of cross-alignment score computation, using the state-of-the-art WCA scoring method as a representative example. We observe from the heatmap that the WCA score corresponding to the ground-truth label becomes more dominant after removing redundant image patches through a simple *Intersection over Union (IoU)* filter. Since the WCA score is computed by aggregating similarity scores between image patches and the textual description, redundant patches introduce repetitive contributions that accumulate disproportionately in the final score. By contrast, after applying IoU filter, the resulting WCA score distribution better aligns with the correct prediction. Furthermore, we find that as the IoU constraint becomes stricter (e.g., IoU = 0.5), the accuracy of the WCA score improves correspondingly. These findings suggest that randomly cropping image patches without enforcing overlap constraints may be suboptimal for accurately computing the cross-alignment score. Hence, these observations motivate us to remove redundant information within these image patches and textual descriptions.

To this end, we propose ***Bi**-refinement for **F**ine-grained **T**ext-visual **A**lignment* (BiFTA), a new method to tackle the above-mentioned issue from two perspectives: *view refinement* and *description refinement*. Specifically, VR uses IoU as the filter metric to efficiently identify and eliminate redundant cropped image patches based on their overlaps of the bounding box. We aim to remove image patches with high IoU ratios, making the remaining visual samples more distinctive. In contrast, *Description Refinement (DR)* first computes the pairwise cosine similarity of the textual descriptions at the representation level, aiming to filter out redundant ones. Then, we select top-$k$ textual descriptions that have the highest cosine similarities with the label caption ("a photo of a/an {label}") from the remaining ones.

Through extensive evaluations across six benchmark datasets, we show that BiFTA outperforms baseline methods by notably improving the zero-shot classification accuracy for both ViT-based and ResNet-based CLIP, justifying the necessity to remove redundant information in visual-text alignment. We summarize the main contributions of our work as follows:

- We observe that localized image patches and fine-grained textual descriptions often contain redundant information, making current SOTA visual-text cross-alignment methods less effective.

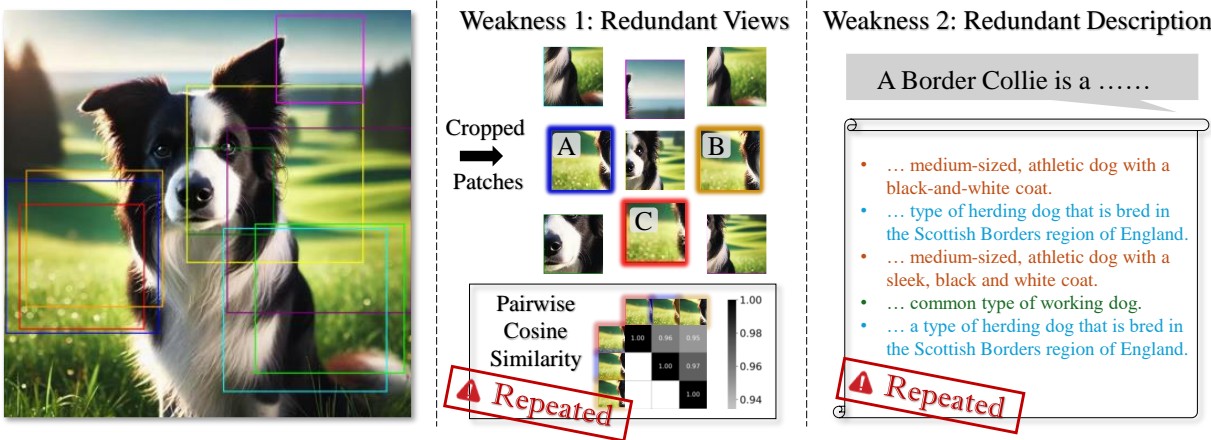

Figure 1: Weaknesses of weighted visual-text cross alignment (Li et al., 2024). **Weakness 1: Pairwise similarity scores of highly overlapping crop bounding boxes.** We demonstrate that image patches A, B, and C, exhibiting significant overlap and redundancy, which provide limited semantic information and consequently contribute minimally to accurate classification. **Weakness 2: Redundant textual descriptions generated by LLM.** We gather textual descriptions from previous work and demonstrate that a significant portion of these descriptions are redundant for a given category, thereby diluting the contribution of meaningful and informative descriptions.

- We propose an efficient data refinement method, namely BiFTA, to mitigate such redundancy through VR and DR, enhancing the alignment between visual and textual modalities.

- We empirically show that BiFTA outperforms baseline methods by achieving significant improvements in zero-shot classification accuracy across 6 benchmark datasets with various CLIP backbones.

## 2 Related Work

**Zero-shot learning for Vision-Language Models.** *Vision-language models* (VLMs) have shown their emergent capabilities on image captioning, visual question answering and image classification, which are not specifically pre-trained or explicitly finetuned for these downstream tasks (Radford et al., 2021; Cho et al., 2021; Wang et al., 2021; Kim et al., 2021; Xue et al., 2021; Li et al., 2022a; Alayrac et al., 2022; Chen et al., 2025; Cai et al., 2025c). CLIP (Radford et al., 2021) demonstrates that integrating large-scale pre-training on image-text pairs with a contrastive loss function could enable zero-shot transfer to downstream tasks by simply using natural language prompts. Similarly, ALIGN (Jia et al., 2021) further demonstrates robust representation learning capability of VLMs by pre-training on noisy image-text pairs at large scale. By increasing the scale of the pre-training data and model size, Florence (Yuan et al., 2021) introduces a unified vision-language foundation model capable of zero-shot image classification and retrieval. On the other hand, CoCa (Yu et al., 2022) combines contrastive and generative objectives to improve zero-shot generalization across diverse tasks. The scaling of pre-training data and the contrastive learning paradigm contribute to deeper visual-text alignment and visual understanding of the model.

**Textual prompt engineering in VLMs.** By scaling the training data, VLMs can learn and understand diverse visual concepts, which can then be transferred to downstream tasks through specific textual label prompting (Radford et al., 2021; Jia et al., 2021; Yuan et al., 2021; Li & Liang, 2021; Singh et al., 2022; Zhou et al., 2022; Shu et al., 2022; Cai et al., 2024b;a; Cui et al., 2025; Ye et al., 2025; Yin et al., 2025). The LLM-integrated textual description generation shows a great generalization ability comparing with existing prompt-learning methods, which often overfit to training data (Li et al., 2022b; Wang et al., 2022; Wu et al., 2023; Tanwisuth et al., 2023; Cai et al., 2025b). CLIP (Radford et al., 2021) achieves zero-shot classification

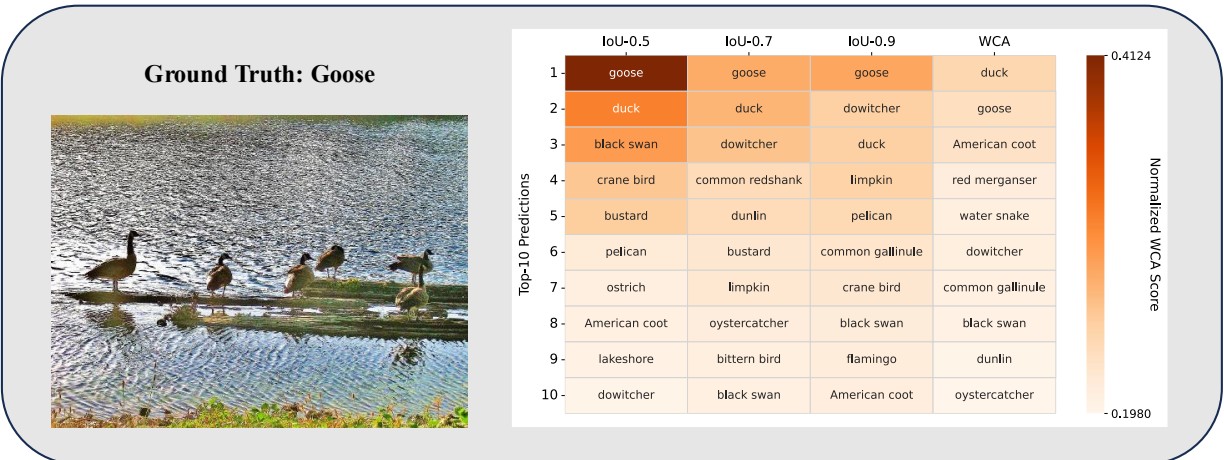

Figure 2: We select an ImageNet image of a goose and display the Top-10 predictions ranked by WCA scores. The scores are normalized using softmax and its distribution is visualized using color intensity. The results show that applying an IoU-based filter to eliminate duplicated image patches significantly enhances the precision of WCA scoring.

by generating classification weights through encoding textual descriptions that uses CLIP template and categories via its text encoder. It then compares these text embeddings with image features extracted by the image encoder to determine the most likely class. Zhou et al. (2022) discover that manually prompt tuning is a time-consuming task and propose CoOp, which models context words with continuous vectors. Subsequently, Menon & Vondrick (2023); Pratt et al. (2023) automatically generates textual category-specific descriptions by leveraging LLMs with different prompt templates. These textual descriptions can accurately reflect visual features of images in each category. More recently, *retrieval-augmented generation* (RAG) is proposed to help to generate accurate descriptions of categories, which is a training-free framework that can be directly integrated during inference time (Yu et al., 2024; Chan et al., 2024; Guo et al., 2024). It retrieves semantically relevant documents by computing embedding vector similarity and provides the retrieved information as supplementary context to LLMs, enabling more accurate and informed responses.

**Fine-grained visual-text alignment.** *Weighted visual-text cross alignment* (WCA) (Li et al., 2024) has done empirical observation on the embedding alignment between visual patches and textual descriptions. It uses random crops to augment image samples and utilizes cosine similarity to extract informative patches. Similarly, it utilizes distinctive textual descriptions from LLM to cross-align with the image patches. There are uncertainties when cropping samples randomly, and the textual descriptions are not corresponding to fine-grained image patches. AttrVR (Cai et al., 2025a) uses descriptive and distinctive attributes of each categories from LLM outputs. In our method, we augment the textual descriptions by integrating various description generation methods to further select the high-quality textual descriptions.

## 3 Preliminary

**CLIP Zero-shot Classification.** CLIP (Radford et al., 2021) is a pre-trained VLM that consists of a text encoder $f_{\text{txt}} : \mathcal{T} \to \mathcal{Z}$ and an image encoder $f_{\text{img}} : \mathcal{X} \to \mathcal{Z}$, where $\mathcal{T}$ is a discrete text space, $\mathcal{X}$ is a continuous image space and $\mathcal{Z} \subseteq \mathbb{R}^n$ is a shared $n$-dimensional embedding space. These encoders take an image $X \in \mathcal{X}$ and a text $T \in \mathcal{T}$ as input pair $(X, T)$, mapping them into the shared latent space $\mathcal{Z}$. Then the similarity score between the image and text embedding is calculated as:

$$\text{sim}_{\text{CLIP}}(X, T) = \cos(Z_i, Z_t)/\tau, \text{ with } Z_i = f_{\text{img}}(X), \text{ and } Z_t = f_{\text{txt}}(T),$$

where $\cos(\cdot, \cdot)$ denotes the cosine similarity such that $\cos(Z_i, Z_t) = \frac{Z_i \cdot Z_t}{\|Z_i\| \|Z_t\|}$ and $\tau$ is a temperature parameter.

For downstream classification tasks, the text encoder of CLIP model receives a label prompt string $\hat{T}_y$ (e.g., "This is a photo of a/an $[y]$"), where $y \in \mathcal{Y}$. Subsequently, CLIP model predicts the label which maximizes the probability of $p_{\text{CLIP}}(Y \mid X)$, given by:

$$\underset{y \in \mathcal{Y}}{\arg\max}\, p_{\text{CLIP}}(Y = y \mid X) = \frac{\exp\!\big(\text{sim}_{\text{CLIP}}(X, \hat{T}_y)\big)}{\sum_{y' \in \mathcal{Y}} \exp\!\big(\text{sim}_{\text{CLIP}}(X, \hat{T}_{y'})\big)}.$$

Here, the label $y$ that maximizes the conditional probability $p_{\text{CLIP}}(Y = y \mid X)$ will be chosen, where the visual and textual representations exhibit the highest similarity within the embedding space $\mathcal{Z}$. This enables zero-shot classification capability on CLIP model, as it can generalize to unseen categories without additional fine-tuning.

**Weighted Cross-Alignment (WCA).** WCA is a scoring method specifically designed to improve the visual-text cross-alignment capability of the CLIP model (Li et al., 2024). First, an original image $X = x$ is randomly cropped with a window size ranging from $[\alpha, \beta] \in [0, 1]$ for $n$ times. It obtain $n$ cropped image patches denote as $I_i = \text{rnd\_crop}(x)$, $i \in [0, n]$, where $\text{rnd\_crop}(\cdot)$ is the random cropping function that obtains localized visual features (Li et al., 2024). On the other hand, WCA would prepare $m$ textual descriptions $T_1, T_2, \ldots, T_m$ generated by LLMs, which encompass descriptive features of each category $y \in \mathcal{Y}$. The textual descriptions are collected by leveraging a LLM with manually crafted prompt templates, such as "Describe what a/an category looks like." (Pratt et al., 2023). Then the visual-text similarity score matrix can be presented as:

$$\begin{bmatrix} \text{sim}_{\text{CLIP}}(I_1, T_1) & \cdots & \text{sim}_{\text{CLIP}}(I_1, T_m) \\ \vdots & \ddots & \vdots \\ \text{sim}_{\text{CLIP}}(I_n, T_1) & \cdots & \text{sim}_{\text{CLIP}}(I_n, T_m) \end{bmatrix},$$

where $\text{sim}_{\text{CLIP}}(I_i, T_j)$ represents the similarity scores between a specific textual description $T_j$ and an image patches $I_i$. When applying WCA, the overall similarity score between image $x$ and label $y$ is as follows:

$$\text{sim}_{\text{WCA}}(X = x, Y = y) = \sum_{i=1}^{n} \sum_{j=1}^{m} w_i v_j\, \text{sim}_{\text{CLIP}}(I_i, T_j), \tag{1}$$

where $w_i$ and $v_j$ are weights for image patch $I_i$ and textual description $T_j$, respectively. They are obtained from the similarity between $I_i$ and the original image $x$, or $T_j$ and the label prompt string $\hat{T}_y$, i.e., $w_i = \text{softmax}\big(\cos(f_{\text{img}}(x), f_{\text{img}}(I_i))\big)$ and $v_j = \text{softmax}\big(\cos(f_{\text{txt}}(\hat{T}_y), f_{\text{txt}}(T_j))\big)$. WCA effectively aligns visual-text pairs with aforementioned weights (Li et al., 2024). However, redundant information may appear in image patches and textual descriptions, leading to the weaknesses shown in Figure 1.

## 4 BiFTA: View Refinement and Description Refinement

The BiFTA framework seeks to eliminate redundant information from two primary perspectives: (1) VR (Section 4.1), which removes overlapping image patches based on IoU scores; and (2) DR (Section 4.2), which integrates multiple methods for generating fine-grained textual descriptions and filters semantically similar embeddings using cosine similarity scores. An overview of BiFTA is illustrated in Figure 3. Our approach integrates efficient data filtering and refinement techniques to refine cross-aligned image-text pairs, ultimately enhancing the quality and accuracy of image classification. Furthermore, in Section 4.4, we formalize the concept of *Redundant Views/Descriptions* and *BiFTA-deduplicated set*.

### 4.1 View Refinement

As shown in Section 3, in WCA, for a single image $x$, a set of randomly cropped image patches $V = \{I_1, I_2, \ldots, I_n\}$ is created for subsequent cross-alignment. However, as illustrated in Figure 1, the random cropping method frequently selects the same region or adjacent regions for cropping, which could affect the classification results. Therefore, without changing the size of the set $|V|$, we make $|V|$ into a queue to store all the image patches. Then we employ a filtering function $f_{\text{IoU}}(\cdot)$ to ensure that each newly cropped image

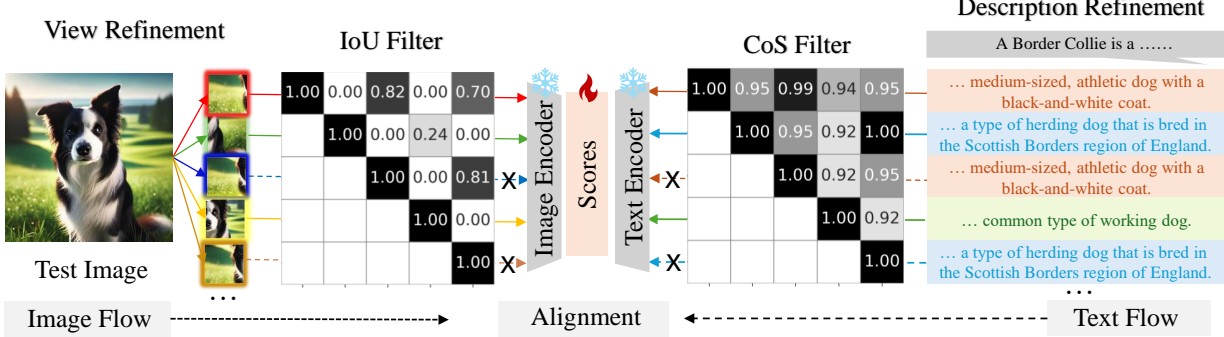

Figure 3: **An Overview of BiFTA.** To reduce potential redundancy in views and descriptions, the randomly cropped views undergo filtering with the IoU filter (Section 4.1), while the randomly sampled description texts are processed using the CoS filter (Section 4.2) when computing the similarity between a single image and a single label. The similarity score is then calculated on the refined views and descriptions.

patch added to the current $V$ does not have excessive overlap areas with the existing images in current queue. For a newly cropped image $I_i = \text{rnd\_crop}(x)$, the filtering function can be expressed as:

$$f_{\text{IoU}}(I_i, V) = \begin{cases} 1 & \forall I \in V, \ \text{IoU}(I, I_i) < 1 - \delta \\ 0 & \exists I \in V, \ \text{IoU}(I, I_i) \geq 1 - \delta \end{cases}, \tag{2}$$

where $\text{IoU}(\cdot)$ is the Jaccard index[1] (i.e., intersection over union) between two image patches, and $1 - \delta$ is a hyperparameter representing the IoU threshold detailed in Section 5. When $f_{\text{IoU}}(I_i, V) = 0$, the view set $V$ remains unchanged. However, if $f_{\text{IoU}}(I_i, V) = 1$, $V$ will be appended with the new view $I_i$, i.e., $V \leftarrow V \cup \{I_i\}$. Thus, by maintaining the size of $V$ consistent with WCA, our method effectively introduces a greater number of semantically independent views.

## 4.2 Description Refinement

In previous works, CuPL (Pratt et al., 2023) used label-integrated prompt templates such as "Describe what a/an [label] looks like" to obtain various appearance descriptions of different classes. Meanwhile, AttrVR (Cai et al., 2025a) employs prompts like "Describe the appearance of [task] [label]" to obtain DesAttr (which describes intra-class features) and DistAttr (which distinguishes inter-class features). Both methods involve manual filtering to remove noise from low-quality generations. As they are built upon LLM-based generation with carefully designed templates, the resulting textual descriptions are semantically aligned and comparable in granularity instead of heterogeneous. Accordingly, we initiate the DR process by taking the union of the two description sets produced by CuPL and AttrVR, treating it as a form of data augmentation. For a given label $y$, let us denote the three sets of descriptions as $T^{\text{CuPL}}(y), T^{\text{Des}}(y), T^{\text{Dist}}(y)$, respectively. We can then formulate the equation as:

$$T^{\text{CuPL}}(y) = f_{\text{LLM}}(y|[\text{cupl\_prompt}]), \ T^{\text{Des}}(y) = f_{\text{LLM}}(y|[\text{des\_prompt}]), \ T^{\text{Dist}}(y) = f_{\text{LLM}}(y|[\text{dist\_prompt}]),$$

where $f_{\text{LLM}}(\cdot)$ returns the LLM output given queries, and $\text{cupl\_prompt}, \text{des\_prompt}, \text{dist\_prompt}$ are aforementioned prompts used by these methods.

Similar to the VR, in order to alleviate the redundancy when obtaining description set $D = \{T_1, T_2, \ldots, T_m\}$, we also employ a filtering function $f_{\text{CoS}}(\cdot, \cdot)$ during random sampling. When sampling text $T_i \in T^{\text{CuPL}}(y) \cup T^{\text{Des}}(y) \cup T^{\text{Dist}}(y)$, the filter function $f_{\text{CoS}}(T_i, D)$–which determines whether new sampled $T_i$ should be included in current set $D$–can be represented as $f_{\text{CoS}}(T_i, D) = f_{\text{CS}}(T_i, D) \cdot f_{\text{TopK}}(T_i, D)$, which consists of a

---

[1]https://en.wikipedia.org/wiki/Jaccard_index

filter $f_{\text{CS}}(\cdot, \cdot)$ that removes similar descriptions:

$$f_{\text{CS}}(T_i, D) = \begin{cases} 1 & \forall T \in D, \ \cos\big(f_{\text{txt}}(T), f_{\text{txt}}(T_i)\big) < 1 - \epsilon \\ 0 & \exists T \in D, \ \cos\big(f_{\text{txt}}(T), f_{\text{txt}}(T_i)\big) \geq 1 - \epsilon \end{cases},$$

and another filter $f_{\text{TopK}}(\cdot, \cdot)$ for eliminating noisy or irrelevant descriptions:

$$f_{\text{TopK}}(T_i, D) = \begin{cases} 1 & T_i \in \text{Top-}k\big(\cos(f_{\text{txt}}(\hat{T}_y), f_{\text{txt}}(T)) \mid T \in D\big) \\ 0 & T_i \notin \text{Top-}k\big(\cos(f_{\text{txt}}(\hat{T}_y), f_{\text{txt}}(T)) \mid T \in D\big) \end{cases},$$

where $\epsilon$ is a hyperparameter representing the threshold, Top-$k(\cdot)$ returns the set of variable $T$s corresponding to the top $k$ function values, and $\hat{T}_y$ is the label prompt string (e.g. "This is a photo of a/an [label]").

The use of $f_{\text{CS}}(\cdot, \cdot)$ ensures that the selected description set contains as little repetitive or redundant textual content as possible. The use of $f_{\text{TopK}}(\cdot, \cdot)$ minimizes the presence of distracting descriptions in the candidate description set (e.g., noisy text generated by an LLM). When $f_{\text{CoS}}(T_i, D) = 0$, the description set $D$ remains unchanged. However, if $f_{\text{CoS}}(T_i, D) = 1$, $D$ will be appended with the new description $T_i$, i.e., $D \leftarrow D \cup \{T_i\}$.

In conclusion, DR first forms a unified description pool by combining two high-quality description sets, then removes duplicate pairs and only keeps the top-k semantically matching pieces into our description pool. As a result, the textual descriptions become more diverse under fixed set size $|D|$, improving the effective utilization of informative descriptions.

### 4.3 Overall Pipeline

The overall pipeline of BiFTA is illustrated in Figure 3. From the Image Flow illustrated in the figure, BiFTA performs VR to the randomly cropped image patches. Each patch is stored in a fixed size queue $V$, while each image patch is filtered using the IoU filter $f_{\text{IoU}}$, as defined in Eq. 2. From the Text Flow illustrated on the other side of the figure, BiFTA conducts DR on a set of textual descriptions. The candidate texts are first filtered by the CoS filter $f_{\text{CoS}}$ and then ranked according to their top-$k$ similarity scores with the label prompt. Finally, the similarity between the refined views and descriptions is computed using Eq. 1 to yield the final prediction. The detailed algorithm is provided in Appendix A.

### 4.4 Concept Formalization

In this section, we formally define the concept of *Redundant Views/Descriptions* in Definition 1 and the *Deduplicated Set* in Definition 2.

**Definition 1.** *(Redundant Views/Descriptions). Assuming $I_i$ and $I_j$ are two views of image $x$, if $\text{IoU}(I_i, I_j) \geq 1 - \delta$, where $1 - \delta$ is the IoU threshold, then $I_i$ and $I_j$ are considered to have significant overlap and are regarded as redundant views of each other. Similarly, for textual descriptions $T_p$ and $T_q$ associated with label $y$, if*

$$\cos\big(f_{\text{txt}}(T_p), f_{\text{txt}}(T_q)\big) \geq 1 - \epsilon,$$

*where $1 - \epsilon$ is the cosine similarity threshold, then the two descriptions are considered nearly identical and thus mutually redundant.*

**Definition 2.** *(BiFTA-Deduplicated Set). Given a set of views $V = \{I_1, I_2, \ldots, I_n\}$ of size $n$, $V$ is a deduplicated view set if and only if*

$$\forall\, I_i, I_j \in V, \quad \text{IoU}(I_i, I_j) < 1 - \delta.$$

*Similarly, for a set of textual descriptions $D = \{T_1, T_2, \ldots, T_m\}$ containing $m$ elements, $D$ is a deduplicated description set if and only if*

$$\forall\, T_p, T_q \in D, \quad \cos\big(f_{\text{txt}}(T_p), f_{\text{txt}}(T_q)\big) < 1 - \epsilon.$$

Through Definition 2 and Sections 4.1–4.3, we conclude that the view and description sets used by BiFTA satisfy the deduplicated set constraints. In contrast, the view and description sets employed by WCA do not meet these constraints, as no explicit deduplication is enforced during random image cropping or description sampling.

## 5 Experiment

First, we outline the complete experimental settings in Section 5.1. In summary, we evaluate the refinement performance of BiFTA through extensive experiments on 6 benchmark datasets and 5 different backbone architectures of the CLIP model. To further demonstrate the generalizability of our framework, we additionally conduct experiments on other VLM architectures in Appendix B, including ALIGN (Jia et al., 2021), AltLIP (Chen et al., 2022), GroupViT (Xu et al., 2022), and SigLIP (Zhai et al., 2023), which suggest that BiFTA consistently improves cross-alignment methods. In Section 5.2, we primarily present the zero-shot classification performance of the CLIP model with a ViT-B/32 backbone across 6 downstream tasks. The rest experimental results for other CLIP backbones are provided in Tables 11-13 in Appendix B. In Section 5.3, we conduct ablation studies to validate the design principles of VR and DR. For completeness, we further explore alternative refinement strategies in Appendix E. Finally, the limitations of our work are discussed in Appendix G.

### 5.1 Experimental Settings

**Datasets.** To evaluate BiFTA, we conduct experiments on 6 downstream classification tasks under a zero-shot setting, including: (1) ImageNet (Deng et al., 2009), a large-scale dataset comprising 1,000 diverse object classes; (2) CUB (Welinder et al., 2010), a fine-grained dataset of 200 bird species, focusing on subtle visual distinctions; (3) Oxford Pets (Parkhi et al., 2012), a dataset of 37 pet categories; (4) DTD (Cimpoi et al., 2014), a texture dataset containing 47 categories of materials and surfaces; (5) Food101 (Bossard et al., 2014), a dataset of 101 food categories; and (6) Place365 (Zhou et al., 2017), a scene recognition dataset with 365 categories of indoor and outdoor environments. These datasets span a wide range of categories, which encompass various visual domains such as scenes, textures, food, animals and fine-grained objects. Thus, the evaluation ensures the robustness and generalizability of BiFTA across various real-world applications.

**Baselines.** We evaluate the performance of BiFTA by comparing it with 6 competitive baselines on zero-shot classification: (1) CLIP (Radford et al., 2021), a naive approach that incorporates a manually crafted label prompt; (2) Ensemble CLIP (CLIP-E) (Radford et al., 2021), an advance approach that incorporates a series of label prompts; (3) CLIP-D (Menon & Vondrick, 2023), an approach that utilizes category descriptions generated by a LLM instead of label prompting approach; (4) Waffle (Roth et al., 2023), a novel approach that replaces LLM-generated category descriptions with random word descriptions; (5) CuPL (Pratt et al., 2023), a method that leverages LLM and improves the quality and variety of textual descriptions compared with CLIP-D; (6) WCA (Li et al., 2024), a recently proposed method that computes cross-alignment scores between localized image patches and fine-grained textual descriptions.

**Implementation Details.** We primarily use the pre-trained CLIP model for main experiments, including both *Vision Transformer* (ViT) and ResNet backbone architectures, specifically ViT-B/32, ViT-B/16, ViT-L/14, RN-50 and RN-101. These architectures are selected to enable a thorough analysis of the proposed method across varying scales and complexities. We keep the shared hyperparameters consistent with the settings in WCA (Li et al., 2024): we use $n = 60$ for the patch queue length $|V|$ and $m = 50$ for the number of textual descriptions per category. We keep the cropping window size consistent with WCA, ranging from $[\alpha_{\mathbf{low}}, \beta_{\mathbf{high}}]$, where $\alpha_{\mathbf{low}} = 0.5$ and $\beta_{\mathbf{high}} = 0.9$ across all experiments. Additionally, we store the embeddings of localized image patches during the initial execution (Li et al., 2024), which allows them to be reused when evaluating different sets of textual descriptions, thereby significantly reducing computational costs.

### 5.2 Zero-shot Classification Results

Tables 1 to 2 present the zero-shot classification results across six downstream tasks, with each table corresponding to a different CLIP backbone architecture.

In Table 1, the classification performance of CLIP (B/32) underscores the consistent superiority of BiFTA over all baselines. In specific, BiFTA achieves a **3.33**% improvement in accuracy over the WCA scoring method on the DTD dataset, which consists of diverse texture patterns. This improvement stems from the inherent challenge of the dataset: textual patterns in DTD are often homogeneous, where random cropping strategy tends to produce semantically similar regions. In BiFTA, VR systematically eliminates redundant image

Table 1: Zero-shot classification accuracy (%) across various baseline methods with the pre-trained CLIP model (ViT-B/32). We report the averaged results and standard deviations $\sigma$ of 20 runs, with the improvement $\Delta$(%) over the top-performing baseline WCA highlighted in **green**. The results of our method are highlighted and we use **bold** to represent the best-performing method.

| Method | ImageNet | CUB | Oxford Pets | DTD | Food101 | Place365 |
|---|---|---|---|---|---|---|
| CLIP | 62.05 | 51.21 | 85.04 | 42.93 | 82.60 | 38.51 |
| CLIP-E | 63.37 | 52.74 | 87.38 | 43.83 | 83.93 | 39.28 |
| CLIP-D | 63.01 | 52.69 | 84.46 | 44.20 | 84.12 | 39.90 |
| Waffle | 63.30 | 52.04 | 85.50 | 42.98 | 83.98 | 39.47 |
| CuPL | 64.37 | 49.76 | 87.03 | 47.50 | 84.20 | 39.08 |
| WCA | 66.49 | 56.74 | 89.05 | 49.89 | 86.11 | 40.55 |
| BiFTA (ours) | **66.83**±0.04 | **58.24**±0.17 | **89.74**±0.14 | **53.22**±0.26 | **86.43**±0.06 | **41.55**±0.06 |
| $\Delta$ | **+0.34** | **+1.50** | **+0.69** | **+3.33** | **+0.32** | **+1.00** |

Table 2: Zero-shot classification accuracy (%) across various baseline methods with the pre-trained CLIP model (ViT-L/14). We report the averaged results and standard deviations $\sigma$ of 20 runs, with the improvement $\Delta$(%) over the top-performing baseline WCA highlighted in **green**. The results of our method are highlighted and we use **bold** to represent the best-performing method.

| Method | ImageNet | CUB | Oxford Pets | DTD | Food101 | Place365 |
|---|---|---|---|---|---|---|
| CLIP | 73.48 | 62.12 | 93.24 | 52.61 | 92.55 | 39.63 |
| CLIP-E | 75.52 | 62.53 | 93.62 | 55.43 | 93.07 | 40.55 |
| CLIP-D | 75.03 | 63.26 | 93.30 | 55.05 | 93.03 | 40.55 |
| Waffle | 75.31 | 62.27 | 91.55 | 54.31 | 93.33 | 40.89 |
| CuPL | 76.62 | 62.15 | 94.33 | 60.59 | 93.37 | 40.77 |
| WCA | 77.32 | 65.12 | 94.67 | 61.74 | 93.93 | 42.19 |
| BiFTA (ours) | **77.82**±0.04 | **65.67**±0.13 | **94.96**±0.10 | **62.45**±0.26 | **93.97**±0.04 | **42.98**±0.05 |
| $\Delta$ | **+0.50** | **+0.55** | **+0.29** | **+0.71** | **+0.04** | **+0.79** |

Table 3: The average Zero-shot classification accuracy (%) over five different CLIP backbones across all downstream benchmarks. Improvements are shown in **green**.

| Dataset | WCA | BiFTA | $\Delta$ |
|---|---|---|---|
| ImageNet | 68.10 | **68.91** | **+0.81** |
| CUB | 55.31 | **56.05** | **+0.74** |
| Oxford Pets | 90.34 | **90.38** | **+0.04** |
| DTD | 52.77 | **54.52** | **+1.75** |
| Food101 | 87.04 | **87.18** | **+0.14** |
| Place365 | 40.22 | **41.15** | **+0.93** |

patches through the IoU filter function. The image patch queue $V$ is thereby maintained as a deduplicated set consisting of semantically independent candidates. In Eq. 1, each selected image patch is assigned a corresponding visual weight $w_i$. This weight factor ensures that discriminative visual features dominate the computation of the cross-alignment score, thereby enhancing the model's robustness against repetitive patterns such as the textures in DTD dataset. The text weight factor $v_j$ follows analogously, where it highlights semantically distinctive textual descriptions. In Table 2, we observe that BiFTA consistently outperforms all baseline methods when evaluated with a large-scale backbone (ViT-L/14). Notably, it achieves a (+0.50%) improvement on ImageNet, which is a significant gain for such a high-capacity model. Notably, BiFTA yields only marginal improvements on large-scale datasets, such as the +0.04% gain reported in Table 2. The results

Table 4: Average classification accuracy (%) across various baseline methods with different CLIP models. The improvements Δ(%) over the top-performing baseline (i.e., WCA) are highlighted in **green**. We use **bold** to represent the best-performing method and underlined to represent the second-best method.

| Model Architecture | CLIP | CLIP-E | CLIP-D | Waffle | CuPL | WCA | BiFTA (ours) | Δ |
|---|---|---|---|---|---|---|---|---|
| ViT-B/32 | 60.39 | 61.76 | 61.40 | 61.21 | 62.16 | 64.81 | **66.00** | **+1.19** |
| ViT-B/16 | 63.59 | 64.51 | 64.67 | 64.34 | 66.09 | 67.87 | **68.29** | **+0.42** |
| ViT-L/14 | 68.94 | 70.12 | 69.87 | 69.61 | 71.31 | 72.50 | **72.98** | **+0.48** |
| RN-50 | 56.97 | 58.64 | 58.39 | 57.92 | 60.01 | 62.00 | **62.54** | **+0.54** |
| RN-101 | 59.14 | 60.50 | 59.22 | 58.89 | 59.04 | 61.14 | **62.03** | **+0.89** |

Table 5: Comparison of Top-1 accuracy (%) across alternative VLM architectures under zero-shot classification setting. All models are evaluated on the ImageNet-1K benchmark.

| VLMs | Vanilla | Vanilla_E | Vanilla_D | Waffle | CuPL | WCA | BiFTA |
|---|---|---|---|---|---|---|---|
| ALIGN | 65.24 | 65.79 | 65.08 | 65.22 | 66.24 | 66.61 | **67.15** |
| AltLIP | 73.79 | 74.86 | 74.48 | 74.29 | 75.74 | 76.20 | **76.88** |
| GroupViT | 37.11 | 42.72 | 40.10 | 40.42 | 44.53 | 45.27 | **45.31** |
| SigLIP | 76.18 | 76.22 | 76.51 | 76.10 | 77.04 | 77.40 | **77.74** |

obtained with the other CLIP backbones and alternative VLM architectures are reported in Appendix B. To provide a more holistic assessment of model performance, we further report results averaged over five different CLIP backbones across all downstream benchmarks in Table 3. While the average improvements over WCA on certain large-scale datasets remain modest (e.g., +0.04% on Oxford Pets and +0.14% on Food101), prior work has shown that large-scale, high-diversity benchmarks tend to saturate more quickly than smaller or more homogeneous datasets (e.g., DTD or CUB) so that making incremental performance gains less visually prominent (Menon & Vondrick, 2023; Roth et al., 2023). Overall, the averaged results indicate that our method consistently outperforms the baseline across diverse downstream benchmarks. In Table 4, we report the performance of each CLIP backbone averaged over all evaluated benchmarks to provide a complementary view of the results. In specific, BiFTA achieves an average improvement of 0.42% to 1.19% across different CLIP backbones compared to WCA. BiFTA exhibits similar trend observed in original WCA that smaller backbone models (e.g., ViT-B/32) exhibit more significant improvements (+1.19%) comparing with their larger counterparts (e.g., ViT-L/14), (+0.48%). This trend highlights BiFTA's ability to greater enhance weaker backbone representations, leading to relatively larger gains on smaller models.

In Table 5, we report a supplementary experiment that evaluates the performance of BiFTA under alternative VLM architectures. This consistency underscores the adaptability and effectiveness of our refinement method when applied to a diverse range of VLMs. For ease of distinguishing VLM names from method names, we denote CLIP and its two variants (CLIP-E and CLIP-D) as `Vanilla`, `Vanilla_E`, and `Vanilla_D` in Table 5.

For a clearer visual comparison, Figure 4 illustrates the differences between BiFTA and WCA. On the left, given an image of a goose, BiFTA successfully filters out semantically independent regions (e.g., head, wings, neck, and fur), whereas WCA usually produces overlapping image patches (e.g., repeated neck regions or background noise) so that ignores valuable features. This highlights how VR mitigates feature redundancy through IoU-guided filtering. On the right, text samples describing a goose are presented. Descriptions refined by the CoS Filter are notably more diverse and distinctive: semantically meaningful prompts such as "plump, elongated body" and "a long, slender body" are preserved, while redundant phrases like "long necks and webbed feet" are pruned. This demonstrates the effectiveness of DR in promoting semantic diversity among textual inputs.

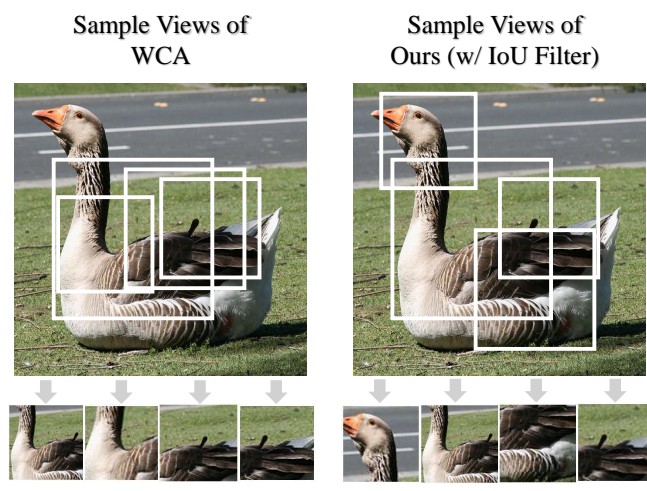

Figure 4: A visualization comparing the effectiveness of VR and DR in BiFTA against WCA. **Left**: with an IoU filter, the cropped samples exhibit diverse and distinctive localized features. **Right**: with a CoS filter, the texts can describe various local features of a category.

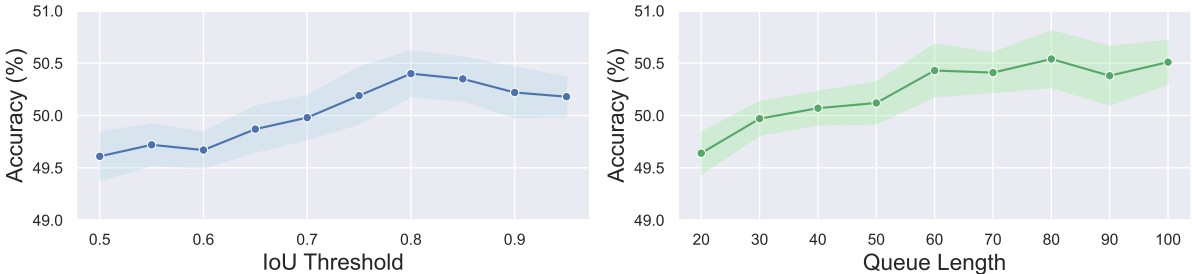

Figure 5: **Left**: Accuracy of using alternative IoU thresholds on DTD dataset with CLIP (B/32). **Right**: Accuracy of changing the size of $|V|$ on DTD dataset with CLIP (B/32).

## 5.3 Hyperparameter Analysis and Ablation Studies

**VR hyperparameter tuning.** Figure 5 illustrates the impact of varying hyperparameter values on downstream performance: IoU threshold $\eta = 1 - \delta$ and patch queue length $N = |V|$. On the left, the classification accuracy varies as the $\eta$ changes, with accuracy gradually rising and then dropping as $\eta$ increases, which suggest that a moderate $\eta$ is needed to achieve an optimal performance. In practice, a lower $\eta$ will result in a critic cut-off to the cropped image patches, which often results in the deduplicated queue $V$ containing an insufficient number of samples. To simply address this limitation, we re-sample from the existing queue until $N$ samples are obtained. However, this procedure inevitably introduces redundant patches into $V$, thereby violating the principle of maintaining a deduplicated set. As shown in the left panel of Figure 5, this trend further indicates that redundant information can adversely affect downstream task performance. Notably, VR becomes ineffective as $\eta$ approaches 1 due to the overly loose restriction on the cut-off. The performance exhibits a decreasing trend once IoU exceeds 0.80, which is expected since the views are no longer effectively constrained by our VR. The right panel of Figure 5 shows that increasing the queue length $N$ has a consistently positive impact on classification accuracy, whereas the increasing trend reaches a plateau at $N = 60$. For consistency, we set $\eta = 0.80$ and $N = 60$ across all experiments.

**DR hyperparameter tuning.** We conduct an ablation study on the cosine similarity threshold $\epsilon$ and the Top-$k$ selection strategy used in DR. Using the ViT-B/32 CLIP model on ImageNet-1K, we evaluate zero-shot performance under different combinations of $(\epsilon, k)$ pairs. Rather than varying a single parameter

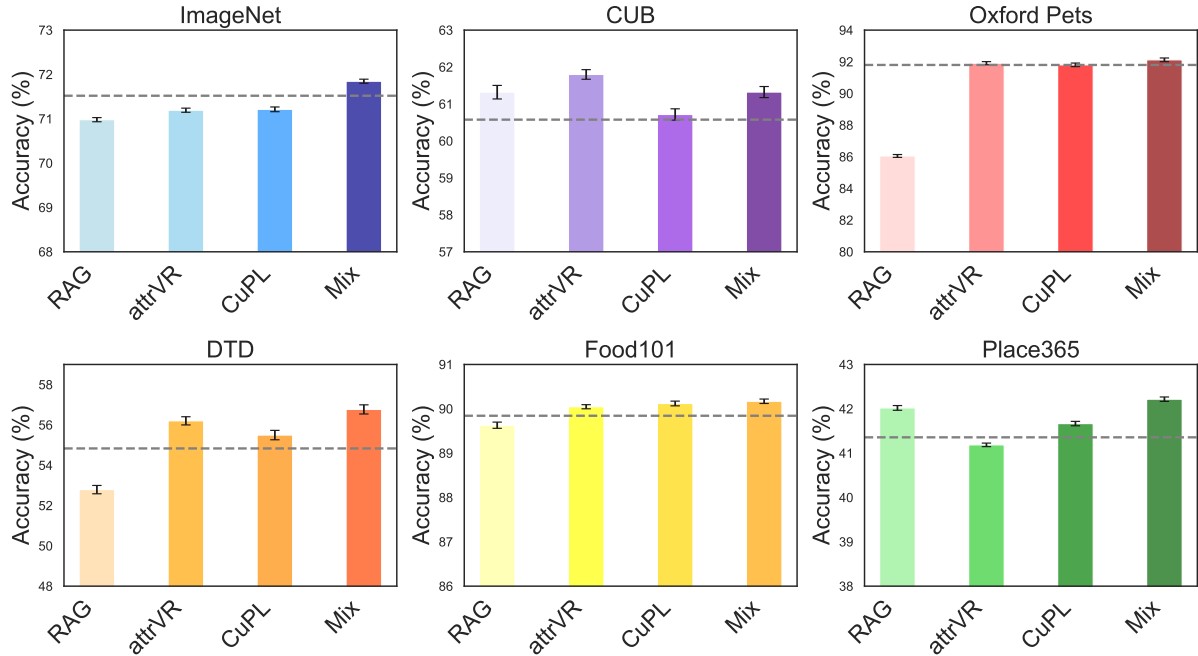

Figure 6: Comparing VR with different description sets and strategies on BiFTA. The results are averaged across 3 image encoder backbones of the CLIP model, where 'mix' is the strategy BiFTA finally utilized for VR.

Table 6: Ablation study on the DR hyperparameters: cosine threshold $\epsilon$ and Top-$k$ selection using ViT-B/32 on ImageNet-1K.

| $(\epsilon, k)$ | w/ DR Acc. (%) |
|---|---|
| $(0.90, 10)$ | $65.02\pm0.07$ |
| $(0.95, 20)$ | $66.45\pm0.04$ |
| $(0.99, 30)$ | $66.70\pm0.04$ |
| $(0.99, 40)$ | $\mathbf{66.84\pm0.05}$ |
| $(0.99, 50)$ | $66.83\pm0.04$ |

independently, we consider $(\epsilon, k)$ pairs, as a strict cosine threshold alone may leave an insufficient number of textual descriptions in the pool. For all main experiments, we adopt $(\epsilon = 0.99, k = 50)$, as specified in Section 5.1. In Table 6, the ablation results show that retaining a larger number of textual descriptions generally leads to improved performance, underscoring the importance of preserving sufficient semantic diversity. In contrast, overly restrictive cosine thresholds discard a substantial portion of candidate descriptions, resulting in reduced diversity and degraded performance. Overall, the setting $(\epsilon = 0.99, k = 50)$ provides a favorable balance between diversity and redundancy, which yields stable and strong performance across benchmarks.

**Ablation studies of alternative DR strategies.** Figure 6 presents the ablation results of using different sets of textual descriptions. These descriptions are generated by LLMs with various prompt template designs, as introduced in Section 4.2. Among all settings, only the mixed description set from our DR demonstrates consistently superior performance across most downstream tasks, with results averaged over three CLIP backbones: ViT-B/32, ViT-B/16, and ViT-L/14. The dashed line indicates the average performance of WCA, which is the strongest baseline method. Relatively, BiFTA achieves varying degrees of improvement depending on the description set. Notably, descriptions generated with RAG-prompt templates perform unsatisfactorily compared with other sets, and details of their implementation are provided in Appendix C. We provide more ablation results of using different sets of textual descriptions in Appendix D. In addition,

Table 7: Ablation studies of comparing the performance of WCA and BiFTA between single modality refinement (w/o VR and w/o DR) and full refinements, using CLIP models (B/32, B/16 and L/14). **VR**: View Refinement; **DR**: Description Refinement. The best result for a single dataset across each model is underlined, and the best averaged results (%) are highlighted in **bold**.

| | Methods | ImageNet | CUB | Oxford Pets | DTD | Food101 | Place365 | Avg. |
|---|---|---|---|---|---|---|---|---|
| B/32 | WCA | 66.49 | 56.74 | 89.05 | 49.89 | 86.11 | 40.55 | 64.81 |
| | BiFTA (w/o VR) | 66.51 | 58.11 | 88.56 | 51.27 | 86.41 | 41.80 | 65.44 |
| | BiFTA (w/o DR) | 66.77 | 56.94 | 89.17 | 50.51 | 87.46 | 40.85 | 65.28 |
| | BiFTA (ours) | 66.83 | 58.24 | 89.74 | 53.22 | 86.43 | 41.55 | **66.00** |
| B/16 | WCA | 71.05 | 59.87 | 92.13 | 52.87 | 89.99 | 41.33 | 67.87 |
| | BiFTA (w/o VR) | 70.67 | 59.36 | 90.58 | 51.36 | 90.20 | 42.23 | 67.40 |
| | BiFTA (w/o DR) | 71.10 | 59.91 | 91.83 | 53.56 | 90.38 | 42.11 | 68.15 |
| | BiFTA (ours) | 71.14 | 60.06 | 91.67 | 54.64 | 90.11 | 42.12 | **68.29** |
| L/14 | WCA | 77.32 | 65.12 | 94.67 | 61.74 | 93.93 | 42.19 | 72.50 |
| | BiFTA (w/o VR) | 77.14 | 65.46 | 94.63 | 62.09 | 93.94 | 42.51 | 72.63 |
| | BiFTA (w/o DR) | 77.89 | 65.56 | 94.78 | 62.17 | 94.04 | 42.58 | 72.84 |
| | BiFTA (ours) | 77.82 | 65.67 | 94.96 | 62.45 | 93.97 | 42.98 | **72.98** |

CuPL is equivalent to applying only the VR component of BiFTA, we also provide quantitative result in Table 7. This suggests that even refining a single modality (the visual side) is sufficient to improve zero-shot image classification performance.

**Ablation studies of BiFTA w/o VR and DR.** We also compare the experimental results of the WCA scoring method with a complete version of BiFTA and partial BiFTA in Table 7. We demonstrate that applying BiFTA refinement to a single modality is sufficient to improve cross-alignment performance. For ImageNet and Food101 datasets, the models often exhibit better performance with BiFTA w/o DR, which indicates our merged description set might not be an optimal description set. Overall, BiFTA with dual refinements achieves the highest average improvement across all three CLIP backbones, as shown in Table 7.

Table 8: Performance comparison among two alternative *View Refinement* (VR) strategies.

| VR | ImageNet | CUB | Oxford Pets | DTD | Food101 | Place365 | Avg. |
|---|---|---|---|---|---|---|---|
| $f_{\mathrm{IoU}}$ | 66.77 | 56.94 | 89.17 | 50.51 | 87.46 | 40.85 | 65.28 |
| $f_{\mathrm{CLIP}}$ | 66.93 | 55.41 | 88.48 | 52.10 | 88.21 | 40.94 | **65.35** |

Table 9: Performance comparison among two alternative *Description Refinement* (DR) strategies.

| DR | ImageNet | CUB | Oxford Pets | DTD | Food101 | Place365 | Avg. |
|---|---|---|---|---|---|---|---|
| $f_{\mathrm{TF-IDF}}$ | 66.38 | 58.05 | 88.74 | 51.25 | 86.19 | 42.14 | **65.46** |
| $f_{\mathrm{CS}}$ | 66.51 | 58.11 | 88.56 | 51.27 | 86.41 | 41.80 | 65.44 |

$$f_{\mathrm{CLIP}}(I_i, V) = \begin{cases} 1 & \forall I \in V, \ \cos\big(f_{\mathrm{img}}(I), f_{\mathrm{img}}(I_i)\big) < 1 - \epsilon \\ 0 & \exists I \in V, \ \cos\big(f_{\mathrm{img}}(I), f_{\mathrm{img}}(I_i)\big) \geq 1 - \epsilon \end{cases},$$

$$f_{\mathrm{TF\text{-}IDF}}(T_i, D) = \begin{cases} 1 & \forall T \in D, \ \cos\big(f_{\mathrm{emb}}(T), f_{\mathrm{emb}}(T_i)\big) < 1 - \epsilon \\ 0 & \exists T \in D, \ \cos\big(f_{\mathrm{emb}}(T), f_{\mathrm{emb}}(T_i)\big) \geq 1 - \epsilon \end{cases},$$

**Exploring Alternative Refinement Strategies.** Table 8 - 9 illustrate the result of the downstream tasks by utilizing alternative VR/DR strategies. For alternative VR strategies, we first attempt to directly use the

Table 10: Runtime time of WCA and BiFTA.

| Execution per image | Time |
|---|---|
| Generate and encode 100 crops | 226.08 ms $\pm$ 10.07 ms |
| IoU filtering | 20.61 ms $\pm$ 7.77 ms |
| **Total time** | 246.69 ms $\pm$ 14.10 ms |

image encoder of the CLIP model $f_{\text{img}}$ to obtain the embeddings of each image patch. Then we incorporate the same cosine similarity function to remove redundant image patches, as shown in Eq. 3. As a result, $f_{\text{CLIP}}$ offers slight improvements in zero-shot classification on downstream tasks. However, computing image patch embeddings is highly computationally intensive, since each sample must be divided into $N$ patches. To make our framework practical, data refinement procedure should be designed in a lightweight manner that minimizes additional computational overhead. In contrast, $f_{\text{IoU}}$ achieves a favorable balance between efficiency and performance. We additionally explore an alternative VR strategy based on grid cropping instead of random cropping in Appendix E, which is a more computational efficient approach.

For alternative DR strategies, we leverage TF-IDF to encode the text descriptions into text embeddings (Jones, 2004), denoted as $f_{\text{emb}}$. Then, we incorporate $f_{\text{emb}}$ into a compositional function $f_{\text{TF-IDF}}$ to eliminate duplicate text descriptions, as shown in Eq. 3. Similar to the proposed $f_{\text{CoS}}$ in Section 4.2, we obtain the final filtering function $f(T_i, D) = f_{\text{TF-IDF}}(T_i, D) \cdot f_{\text{TopK}}(T_i, D)$. The results indicate that the two DR strategies yield nearly identical performance. Nevertheless, both consistently outperform the baselines. The core of DR lies in leveraging the $f_{\text{TopK}}$ function to select the most semantically relevant descriptions to the label prompt after an initial filtering step with either $f_{\text{CS}}$ or $f_{\text{TF-IDF}}$. Overall, lightweight yet principled refinement strategies are sufficient to yield competitive gains without incurring excessive computational overhead.

### 5.4 Runtime Comparison

BiFTA introduces only minor offline preprocessing costs and no additional inference-time overhead. First, VR adds an IoU-based filtering step to the random-crop procedure, incurring an average of 20.61 ms per image compared to 226.08 ms for crop generation and encoding, as shown in Table 10. The patch embeddings are cached and reused during subsequent inference (Li et al., 2024). Then, DR is performed entirely offline via cosine-similarity filtering and Top-$k$ selection, requiring only an average of 42.36$\pm$8.65 ms per category. Overall, both VR and DR introduce only minor, one-time offline preprocessing costs, and BiFTA incurs no additional inference-time overhead compared to WCA.

## 6 Conclusion

In this work, we identified a critical limitation in existing fine-grained visual-text alignment methods: the presence of redundant information in both localized image patches and LLM-generated textual descriptions. To address this, we propose BiFTA, a novel framework that introduces two key innovations: (1) VR via IoU-based filtering to eliminate spatially overlapping image patches, and (2) DR through cosine similarity thresholding to remove semantically redundant textual descriptions. Our experiments across 6 benchmark datasets demonstrate that BiFTA consistently outperforms state-of-the-art methods in zero-shot classification accuracy over the previous methods. The ablation studies validate the necessity of both components: IoU filtering ensures diverse visual features, while cosine-based text pruning enhances semantic specificity. Owing to its flexible design, BiFTA naturally supports both single-modality and dual-modality refinement, and can be readily applied to broader prompt-learning frameworks beyond WCA. Importantly, the refinement process is decoupled from downstream methodologies, which enables broad applicability across diverse tasks and settings.

## 7 Acknowledgements

YHS, CYC and JCZ are supported by the Melbourne Research Scholarship. ZSY is supported by the Australian Research Council (ARC) with grant number DE240101089. FL is supported by the ARC with grant number DE240101089, LP240100101, DP230101540 and the NSF&CSIRO Responsible AI program with grant number 2303037. This research was supported by The University of Melbourne's Research Computing Services and the Petascale Campus Initiative.

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

# A   Appendix 1: Additional Algorithms for the BiFTA Implementation

---

**Algorithm 1** Zero-Shot Classification Pipeline of BiFTA

---

1: **Input** Query image $I_0 \in \mathbb{R}^{H \times W \times 3}$; labels $y \in \mathcal{Y}$; hyperparameters: a patch queue $Q$, crop count $N$, textual description count $M$, a label prompt $\hat{T}_y$, and pre-trained CLIP model with encoders $f_{\text{img}}(\cdot)$ (image) and $f_{\text{txt}}(\cdot)$ (text).

2: **Initialize** $Q \leftarrow [I_1]$ by Eq. 2

  # Step 1: View Refinement

3: **for** $i = 2$ to $N$ **do**

4:     **Generate** $I_i$ by Eq. 2

5:     **Check Is_Redundant**$(I_i)$ by Algo. 2

6:     **Is_Redundant**$(I_i) ==$ **false**, $Q$.push$(I_i)$

7:     **Compute** $w_i = \text{softmax}(\cos(f_{\text{img}}(I_0), f_{\text{img}}(I_i)))$

8: **end for**

  # Step 2: Description Refinement

9: **for** $y \in \mathcal{Y}$ **do**

10:     **Obtain** $\mathcal{T}^y = \{T_j\}_{j=1}^J$ where $J > M$

11:     **Remove** $T_j \in \mathcal{T}^y$ based on Algo. 3

12:     **Obtain** $\tilde{\mathcal{T}}^y \subseteq \mathcal{T}^y$ from Line (11)

13:     **Initialize** $\hat{T}_y$

14:     **Select top-**$M$ $T$s by $\arg\max_{T \in \tilde{\mathcal{T}}^y} \cos(f_{\text{txt}}(T), f_{\text{txt}}(\hat{T}_y))$

15:     **Compute** $v_j$ for all $\{T_j\}_{j=1}^M$ via $v_j = \text{softmax}(\cos(f_{\text{txt}}(\hat{T}_y), f_{\text{txt}}(T_j)))$

16:     **Compute** $\text{sim}_{\text{WCA}}^y$ via Eq. 1

17: **end for**

18: **Output** $y^* = \arg\max_{y \in \mathcal{Y}} \text{sim}_{\text{WCA}}^y$

---

**Algorithm 2** Implementation of Redundant Image Patch Filtering

---

1: **Input:** A patch queue $Q$ contains all the previous saved image patches, a new cropped image patch $I_i$, an IoU threshold $\eta = 1 - \delta$.

2: **Initialize Is_Redundant ==** **false**

3: **for** $k = 1$ to $|Q|$ **do**

4:     **Compute** $\eta_k = \text{IoU}(I_i, Q[k])$

5:     **if** $\eta_k \geq \eta$ **then**

6:         **Is_Redundant** $\leftarrow$ **true**, **break**

7:     **end if**

8: **end for**

9: **Output: Is_Redundant**

---

**Algorithm 3** Implementation of Redundant Textual Description Filtering

---

1: **Input:** A set of textual descriptions of label $y$, $\mathcal{T}^y = \{T_j\}_{j=1}^J$, where $J$ is the number of descriptions in the merged textual description set; $\tau = 1 - \epsilon$ is the tolerance for the duplicate texts, setting to 1.0, $f_{\text{txt}}$ is a text encoder.

2: **for** $j = 1$ to $J - 1$ **do**

3:     **for** $k = j + 1$ to $J$ **do**

4:         **Compute** $S = \cos(f_{\text{txt}}(T_j), f_{\text{txt}}(T_k))$

5:         **if** $S \geq \tau$ **then**

6:             **Remove** $T_k$ from $\mathcal{T}^y$

7:         **end if**

8:     **end for**

9: **end for**

10: **Obtain** $\tilde{\mathcal{T}}^y = \mathcal{T}^y$

11: **Output:** $\tilde{\mathcal{T}}^y$

---

# B  Appendix 2: More Experimental Results

Table 11: Zero-shot classification accuracy (%) across various baseline methods with the pre-trained CLIP model (ViT-B/16). We report the averaged results and standard deviations $\sigma$ of 20 runs, with the improvement $\Delta$(%) over the top-performing baseline WCA highlighted in **green** and **red**. The results of our method are highlighted and we use **bold** to represent the best-performing method.

| Method | ImageNet | CUB | Oxford Pets | DTD | Food101 | Place365 |
|---|---|---|---|---|---|---|
| CLIP | 66.74 | 56.01 | 88.14 | 42.98 | 88.40 | 39.27 |
| CLIP-E | 68.37 | 56.16 | 89.10 | 45.27 | 88.83 | 40.30 |
| CLIP-D | 68.04 | 57.08 | 87.52 | 46.17 | 88.85 | 40.34 |
| Waffle | 68.12 | 56.89 | 86.51 | 44.68 | 89.06 | 40.76 |
| CuPL | 69.61 | 56.42 | 91.14 | 50.53 | 88.98 | 39.83 |
| WCA | 71.05 | 59.87 | 92.13 | 52.87 | 89.99 | 41.33 |
| BiFTA (ours) | **71.14±0.04** | **60.06±0.15** | **91.67±0.11** | **54.64±0.16** | **90.11±0.05** | **42.12±0.04** |
| $\Delta$ | **+0.09** | **+0.19** | **-0.46** | **+1.77** | **+0.12** | **+0.79** |

Table 12: Zero-shot classification accuracy (%) across various baseline methods with the ResNet-based CLIP model (RN-50). We report the averaged results and standard deviations $\sigma$ of 20 runs, with the improvement $\Delta$(%) over the top-performing baseline WCA highlighted in **green**. The results of our method are highlighted and we use **bold** to represent the best-performing method.

| Method | ImageNet | CUB | Oxford Pets | DTD | Food101 | Place365 |
|---|---|---|---|---|---|---|
| CLIP | 58.15 | 45.67 | 83.65 | 38.67 | 78.62 | 37.04 |
| CLIP-E | 59.82 | 46.58 | 85.66 | 41.22 | 80.82 | 37.73 |
| CLIP-D | 59.62 | 47.76 | 83.70 | 42.23 | 79.92 | 37.13 |
| Waffle | 59.82 | 46.76 | 83.54 | 38.88 | 80.74 | 37.77 |
| CuPL | 61.43 | 47.91 | 87.05 | 47.39 | 80.50 | 37.78 |
| WCA | 62.82 | 50.16 | 88.40 | 49.45 | 81.25 | 38.91 |
| BiFTA (ours) | **63.54±0.06** | **50.43±0.17** | **88.74±0.12** | **51.41±0.22** | **81.39±0.09** | **39.70±0.06** |
| $\Delta$ | **+0.72** | **+0.27** | **+0.34** | **+1.96** | **+0.14** | **+0.79** |

Table 13: Zero-shot classification accuracy (%) across various baseline methods with the ResNet-based CLIP model (RN-101). We report the averaged results and standard deviations $\sigma$ of 20 runs, with the improvement $\Delta$(%) over the top-performing baseline WCA highlighted in **green** and **red**. The results of our method are highlighted and we use **bold** to represent the best-performing method.

| Method | ImageNet | CUB | Oxford Pets | DTD | Food101 | Place365 |
|---|---|---|---|---|---|---|
| CLIP | 61.26 | 49.34 | 84.96 | 40.05 | 82.44 | 36.77 |
| CLIP-E | 62.31 | 49.65 | 86.97 | 43.62 | 83.64 | 37.81 |
| CLIP-D | 60.65 | 50.29 | 82.53 | 42.82 | 83.25 | 35.75 |
| Waffle | 61.25 | 48.05 | 83.70 | 40.05 | 82.48 | 37.83 |
| CuPL | 61.43 | 42.85 | 87.63 | 43.83 | 82.74 | 35.77 |
| WCA | 62.82 | 44.64 | 87.43 | 49.91 | 83.92 | 38.11 |
| BiFTA (ours) | **65.24±0.05** | **45.86±0.12** | **86.81±0.15** | **50.87±0.23** | **84.02±0.05** | **39.40±0.06** |
| $\Delta$ | **+2.42** | **+1.22** | **-0.62** | **+0.96** | **+0.10** | **+1.29** |

Tables 11 - 13 provide the results of utilizing alternative architectures of the CLIP backbone. The results show that BiFTA consistently outperforms baselines on downstream tasks with the ResNet-based backbone models, which evidently shows that our framework is able to generalize on various CLIP style models.

## C  Appendix 3: RAG-based Text Generation

We further explore an alternative prompt design based on *Retrieval-Augmented Generation (RAG)* to enrich textual descriptions with external knowledge. Specifically, we construct a knowledge database from a pre-processed Wikipedia corpus[2], which contains approximately 1.8 million documents. All documents are truncated, tokenized, and encoded into embedding vectors using the `text-embedding-ada-002` model, and subsequently stored in the ChromaDB vector database for semantic retrieval.

Due to practical constraints, we utilize a subset of approximately 150k documents in our experiments, as indexing the entire corpus would require prohibitively long preprocessing time. Given a query prompt, documents are retrieved based on cosine similarity between the prompt embedding and document embeddings. The retrieved documents are then concatenated with the prompt template and fed into a GPT model to generate the final textual descriptions.

However, as shown in Figure 6, this RAG-based description generation strategy yields inferior performance when applied to CLIP-based zero-shot classification. We attribute this behavior to two key factors. First, Wikipedia articles typically contain long and diverse contextual information, while the retrieval queries are short prompt templates, making cosine-similarity-based retrieval less effective at identifying visually relevant content. Second, and more importantly, Wikipedia articles often focus on encyclopedic knowledge—such as historical background, taxonomy, or cultural context—rather than fine-grained visual attributes that are critical for visual discrimination. For instance, the Wikipedia entry for a *Persian cat* primarily discusses breeding history and popularity, but provides limited information about localized visual characteristics.

These observations suggest that simply scaling the size of a generic Wikipedia-based database is unlikely to substantially improve performance under the current retrieval and alignment framework. Instead, a more promising direction lies in designing category-aware or attribute-centric RAG strategies that retrieve documents explicitly aligned with patch-level visual semantics. We view such designs as an important avenue for future work, which would require a systematic investigation of suitable external resources and retrieval mechanisms tailored to visual recognition tasks.

## D  Appendix 4: Extra Ablation Studies

Table 14: The ablation study on $\alpha$. This experiment is evaluated on the ImageNet dataset by leveraging CLIP (B/32). We set the upper bound $\beta$ to 0.9 and report the Top-1 Accuracy (%).

| $\beta = 0.9$ | $\alpha$ | | | | | | | |
|---|---|---|---|---|---|---|---|---|
| | 0.1 | 0.2 | 0.3 | 0.4 | 0.5 | 0.6 | 0.7 | 0.8 |
| WCA | 66.48±0.07 | 66.57±0.07 | 66.61±0.06 | 66.72±0.07 | 66.49±0.07 | 66.85±0.04 | 66.92±0.04 | 66.93±0.05 |
| Ours | 66.93±0.06 | 66.97±0.05 | 67.05±0.08 | 67.24±0.04 | 66.83±0.04 | 66.88±0.04 | 66.61±0.04 | 66.55±0.07 |

Table 15: The ablation study on $\beta$. This experiment is evaluated on the ImageNet dataset by leveraging CLIP (B/32). We set the lower bound $\alpha$ to 0.5 and report the Top-1 Accuracy (%).

| $\alpha = 0.5$ | $\beta$ | | | | |
|---|---|---|---|---|---|
| | 0.6 | 0.7 | 0.8 | 0.9 | 1 |
| WCA | 61.77±0.06 | 63.21±0.05 | 64.45±0.05 | 66.49±0.07 | 66.06±0.08 |
| Ours | 64.40±0.07 | 65.46±0.07 | 65.91±0.05 | 66.83±0.04 | 66.73±0.08 |

Table 14 - 15 compare the Top-1 accuracy (%) of the WCA scoring method and BiFTA when varying the cropping size lower bound $\alpha$ and upper bound $\beta$. In Table 14, the results reveal a significant performance gap when the crop window range is large (e.g., $[0.1, 0.9]$). As $\alpha$ increases, image patches are cropped with larger

---

[2]https://huggingface.co/datasets/Salesforce/wikitext/viewer/wikitext-103-v1

window sizes, which reduces the effectiveness of the view filtering under stricter IoU thresholds. In Table 15, as $\beta$ increases, there are more image patches with larger areas. We observe a consistent upward trend in accuracy as $\beta$ increases for both methods. The performance gap gets more pronounced when $\beta$ is smaller. For example, at $\beta = 0.6$, BiFTA achieves a 2.6% higher classification accuracy than WCA. This suggests that when the cropping windows are smaller, redundant small patches have a more significant negative impact on the weighted scores, whereas this issue is effectively addressed by integrating the VR introduced in BiFTA.

Based on Figure 6 in Section 5.3, we provide supplementary results for each backbone. Figures 7 to 9 present the classification accuracy across all downstream tasks for 4 description sets. Each sub-figure represents the results obtained from CLIP ViT-B/32, ViT-B/16 and ViT-L/14, respectively. In conclusion, the mixed set of descriptions, combining the CuPL and AttrVR descriptions and filtering them based on CoS function and top-$k$ similarities, demonstrates superior performance compared to other description sets.

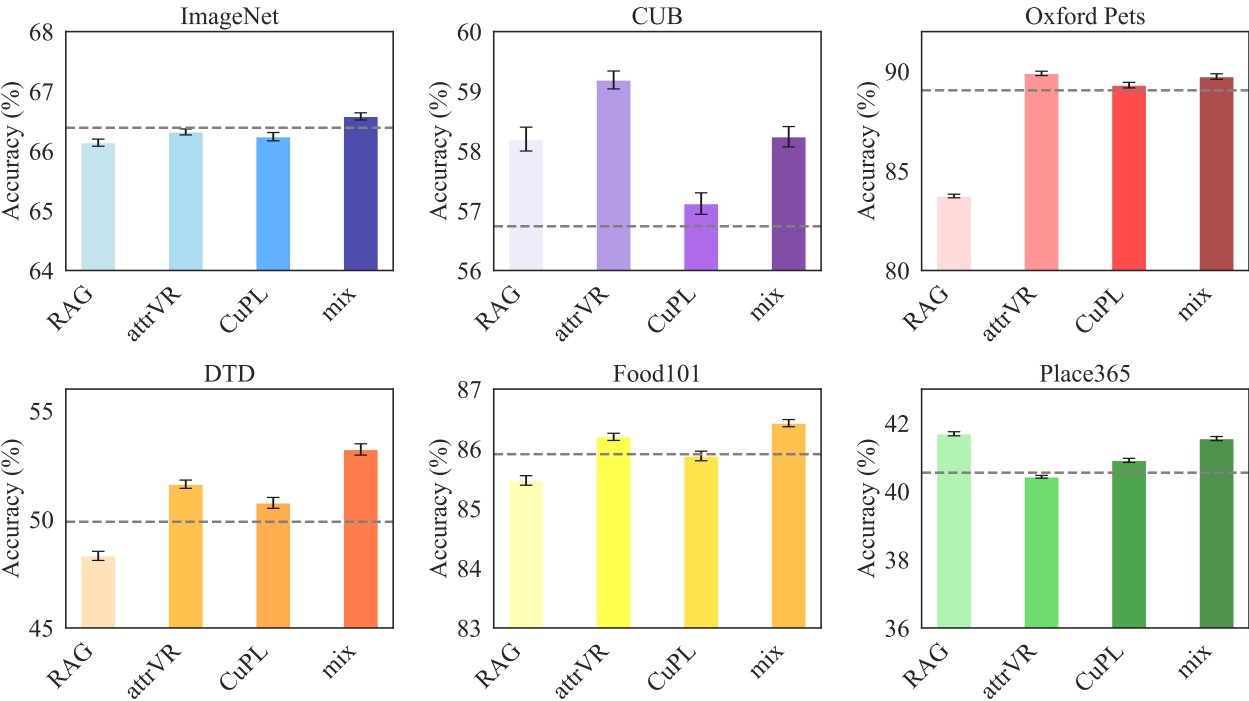

Figure 7: Results of exploring textual description studies, using the results of CLIP (B/32). "RAG" computes similarity using RAG-generated descriptions. "attrVR" uses both descriptive and distinctive texts to calculate similarity. "CuPL" directly employs descriptions from the CuPL method. "mix" combines "attrVR" and "CuPL" descriptions.

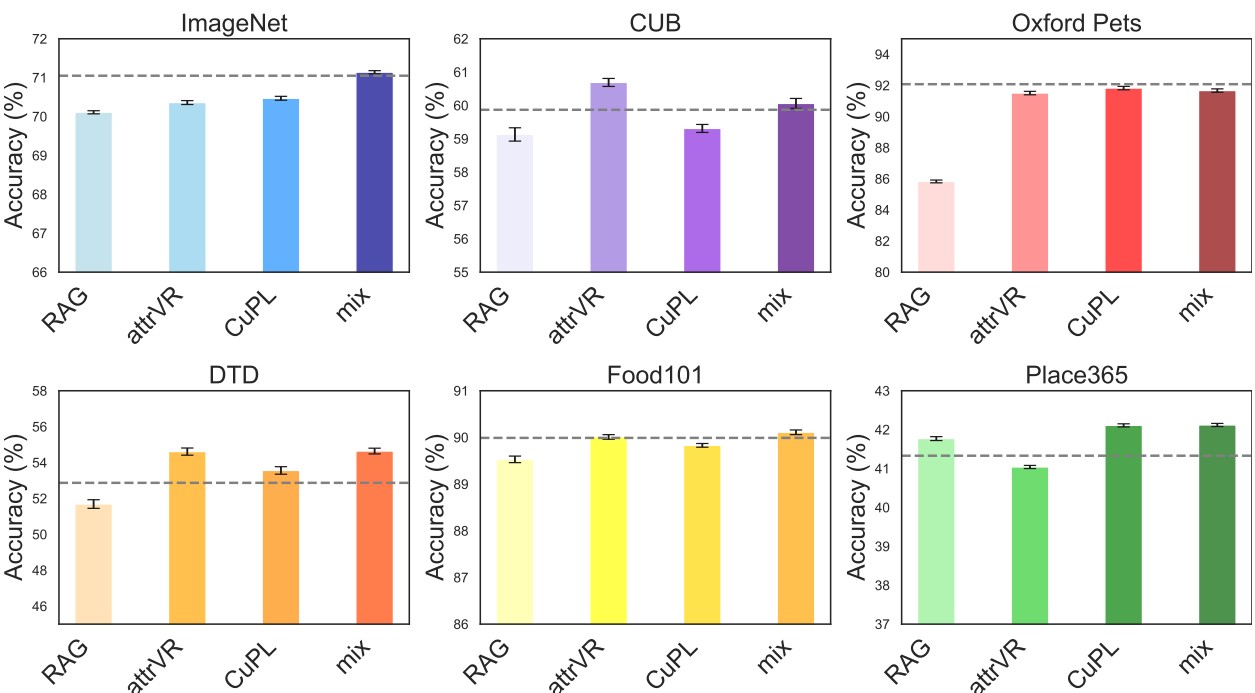

Figure 8: Results of exploring textual description studies, using the results of CLIP (B/16).

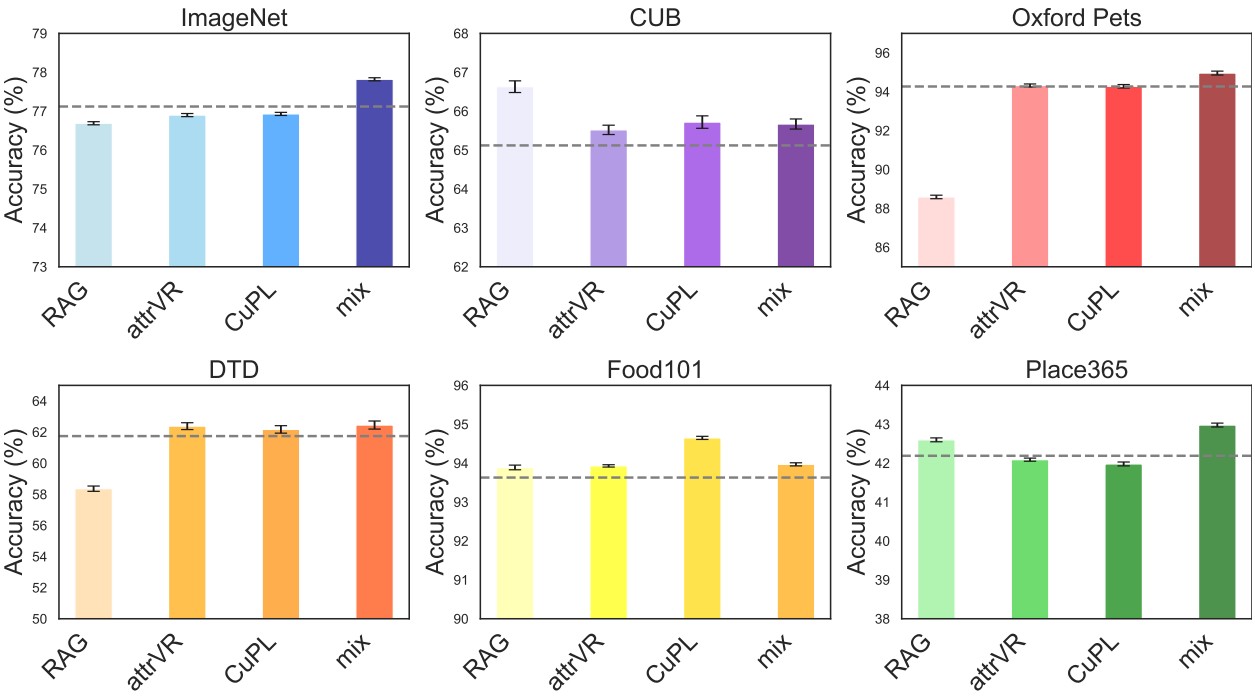

Figure 9: Results of exploring textual description studies, using the results of CLIP (L/14).

# E   Appendix 5: Alternative VR strategy: Grid Crop

Table 16: Zero-shot classification performance of WCA with different grid-crop configurations using ViT-B/32 across multiple benchmarks.

| WCA (ViT-B/32) | CUB | Food101 | DTD | Oxford Pets | ImageNet | Place365 | Avg. acc. |
|---|---|---|---|---|---|---|---|
| Grid crop (3×3) | 36.77±0.15 | 68.09±0.08 | 46.03±0.20 | 76.43±0.18 | 50.67±0.05 | 33.88±0.05 | **51.98** |
| Grid crop (4×4) | 29.18±0.20 | 57.21±0.11 | 43.48±0.28 | 66.60±0.36 | 43.89±0.05 | 31.88±0.07 | 45.37 |
| Grid crop (5×5) | 21.83±0.23 | 45.19±0.09 | 40.19±0.38 | 56.24±0.36 | 37.65±0.11 | 29.34±0.09 | 38.41 |

Table 17: Zero-shot classification performance of VR with different grid-crop configurations using ViT-B/32 across multiple benchmarks.

| VR (ViT-B/32) | CUB | Food101 | DTD | Oxford Pets | ImageNet | Place365 | Avg. acc. |
|---|---|---|---|---|---|---|---|
| Grid crop (3×3) | 38.59±0.22 | 68.70±0.08 | 48.06±0.25 | 77.13±0.23 | 50.92±0.07 | 35.05±0.06 | **53.08** |
| Grid crop (4×4) | 31.65±0.27 | 57.88±0.08 | 44.86±0.25 | 67.73±0.25 | 44.17±0.08 | 33.01±0.06 | 46.55 |
| Grid crop (5×5) | 23.65±0.27 | 46.20±0.18 | 42.28±0.38 | 57.33±0.26 | 37.78±0.08 | 30.47±0.07 | 39.62 |

Compared to approaches that rely on CLIP embedding similarity for patch selection, grid cropping is more computationally efficient, and the resulting cropped patches are inherently non-overlapping. As shown in Tables 16 and 17, BiFTA consistently outperforms WCA under grid-crop-based VR settings. Nevertheless, grid cropping itself leads to noticeable performance degradation compared to random cropping, with classification accuracy consistently decreasing as the grid resolution increases. This trend can be attributed to the reduced patch size induced by finer grid partitioning, where smaller patches contain substantially less semantic information and lead to worse cross alignment. A similar pattern is also observed in Table 14. Overall, while grid cropping serves as a viable alternative VR baseline, these results further demonstrate that BiFTA remains robust and consistently surpasses WCA across diverse VR configurations.

# F   Appendix 6: Utilizing advanced LLM

Table 18: Comparison using textual descriptions generated by a more advanced language model.

| Description set | ImageNet (%) |
|---|---|
| Original | **66.83±0.04** |
| GPT-4o | 66.64±0.04 |

We further investigate the impact of using a stronger language model for description generation by constructing an alternative set of textual descriptions for ImageNet categories using `gpt-4o`[3]. To ensure a fair comparison, we adopt five representative prompt templates from CuPL and AttrVR and generate ten description samples per prompt for each class. The corresponding results are reported in Table 18.

We observe a slight performance degradation compared to the original description set, suggesting that although more advanced language models may produce richer or more detailed descriptions, such improvements do not directly translate into better performance under the current cross-alignment scoring framework. This result indicates that the effectiveness of description refinement is not solely determined by the strength of the language model, but also by how well the generated descriptions align with patch-level visual representations. These findings point to an interesting future direction: designing prompt templates that are explicitly tailored to localized visual features, which may further enhance the effectiveness of description refinement.

---

[3]https://platform.openai.com/docs/models/gpt-4o

# G   Appendix 7: Limitation

One potential limitation we observed is that the textual descriptions generated by the LLM do not consistently focus on local features of the object. These descriptions often tend to be generic, making it difficult to associate specific parts of the text with corresponding local image patches from a human perspective. To address this, we aim to explore more advanced approaches to generate precise and localized descriptive texts that better align with the visual details of an image.

On the other hand, while BiFTA generally benefits from incorporating more diverse visual views and textual descriptions, its effectiveness relies on the assumption that increased patch diversity improves fine-grained cross-modal alignment. In practice, we observe that this assumption may not always hold uniformly across all benchmarks. For instance, on Oxford Pets, BiFTA sometimes exhibits a slight performance drop compared to WCA, suggesting that certain patches filtered as redundant may still contain class-discriminative information. This observation highlights a limitation of the current refinement strategy: it does not explicitly distinguish between truly redundant patches and visually similar yet semantically important ones. Identifying such cases and developing adaptive refinement mechanisms that better preserve informative visual cues remain important directions for future work.

