# OpenReview forum: "Let's Roll a BiFTA: Bi-refinement for Fine-grained Text-visual Alignment in Vision-Language Models"
_TMLR — Accepted by TMLR_

### Review · Reviewer_4Aac · 2025-10-21

**Summary Of Contributions:**

The paper proposes a new method called BiFTA to improve the zero-shot performance of vision–language models (VLMs) such as CLIP. The method consists of two main components: (1) view refinement and (2) description refinement, which remove redundant image patches and redundant or irrelevant textual descriptions, respectively. Experiments on six benchmarks and across different architectures demonstrate the superiority and robustness of the proposed method.

**Audience:**

Yes

**Audience Explanation:**

Although the improvements on some benchmarks are minor, certain readers may be interested in further exploring the boundary of the WCA method—for example, investigating whether its performance could be enhanced by replacing the random cropping strategy with a more structured or deterministic one.

**Claims And Evidence:**

Yes

**Claims Explanation:**

The method is intuitively sound, and the improvement direction is well-motivated. Evaluations across various benchmarks and architectures, along with ablation studies, demonstrate the readiness and effectiveness of the proposed approach.

**Requested Changes:**

As this paper is a resubmission, I notice only a few adjustments:

1. What is the computational efficiency of the proposed method compared with WCA?

2. Is there any ablation study on the patch size and the length of the LLM-generated descriptions?

3. Which LLM is used in the paper to produce the detailed descriptions? Any ablation on the LLM used, e.g. using more advanced LLM to provide better descriptions?

---

> ### Author Response · Authors · 2025-12-06
> **Response to Reviewer 4Aac**
>
> We sincerely thank the reviewer for the encouraging and thoughtful feedback. We address the questions and suggestions in detail below.
>
> **[Q1]**: What is the computational efficiency of the proposed method compared with WCA?
>
> **[R1]**: Since **BiFTA is a lightweight input-refinement framework**, its additional cost arises only during the pre-processing stage rather than at inference time. For VR, BiFTA introduces an IoU-based filtering mechanism into random-crop procedure. Each cropped patch is checked via the IoU function and stored in a patch queue. The measured time cost for one image is reported below, it shows that IoU filtering adds only 20.61 ms, whereas generating and encoding all crops require 226.08 ms per image. Thus, the computational overhead is minimal and does not meaningfully impact the overall computation. Furthermore, the stored image patch embeddings are stored as tensors into pkl files, so that VR only introduces additional inference costs during the first generation of the patch embeddings.
>
> | Execution per image           | time                      |
> |------------------------------|---------------------------|
> | Generate and encode 100 crops | 226.08 ms ± 10.07 ms      |
> | IoU filtering                 | 20.61 ms ± 7.77 ms        |
> | Total time                    | 246.69 ms ± 14.10 ms      |
>
>
> For DR, we clarify that it also does not introduce extra runtime during inference as DR operates entirely offline. We first take the union of two high-quality description sets from CuPL [1] and AttrVR [2], then apply cosine-similarity filtering and Top-K ranking. Each category typically contains around 100 candidate descriptions, and filtering + selecting the Top-50 requires only 42.36 ± 8.65 ms, which is negligible. Finally, we save the filtered textual description sets into JSON files. Just as in WCA, we directly leverage a set of descriptions as textual inputs to the CLIP model during inference time. Overall, both VR and DR incur **only minor, one-time offline preprocessing costs**.
>
> **[Q2]**: Is there any ablation study on the patch size and the length of the LLM-generated descriptions?
>
> **[R2]**: We would like to clarify that our work already includes an ablation study on patch size, and we summarize the key findings: As shown in App. 5 (Tables 11–12), varying the patch size on ImageNet demonstrates that **larger patches consistently lead to higher accuracy**, as they preserve coherent local semantic information. In contrast, very small patches fragment meaningful cues and consequently degrade classification performance.
> For the length of LLM-generated descriptions, we fix the maximum output length to 50 tokens, which is sufficient to capture salient local attributes without introducing excessive global context. Since the cross-alignment score is designed to **match localized image features with corresponding localized textual cues** (Figure 4, Left), overly long descriptions may embed multiple unrelated attributes, which requires truncation and potentially weakening the cross-alignment signal. While conducting a full ablation on description length is feasible, it would require substantial additional computation to report the result later.
>
> **[Q3]**: Which LLM is used in the paper to produce the detailed descriptions? Any ablation on the LLM used, e.g. using more advanced LLM to provide better descriptions?
>
> **[R3]**: Our Description Refinement (DR) pipeline first aggregates two existing high-quality LM-generated description sets from CuPL [1] and AttrVR [2], and then refines them using cosine-similarity filtering and Top-K selection. As detailed in Section 4.2 and in **[R2]** (Reviewer tbw9), CuPL employs OpenAI’s `text-davinci-002` model, while AttrVR uses `gpt-3.5-turbo-instruct` to produce category-specific descriptions. These two sources provide semantically rich, human-interpretable descriptions, making them suitable foundations for refinement.
>
> | Description set   | ImageNet      |
> |-------------------|----------------|
> | original          | 66.83±0.04   |
> | gpt-4o  | 66.64 ± 0.04   |
>
> We additionally generate a set of textual descriptions for ImageNet categories by employing a more advanced `gpt-4o` model. To ensure fairness, we curated five representative prompt templates from CuPL and AttrVR, then produced ten samples per prompt for each class. The updated results are reported below. We observe marginal degradation performance, so that while stronger LMs may yield slightly richer descriptions, the benefits are limited under our current cross-alignment scoring framework. Thus, this experiment highlights a promising direction—**designing prompt templates tailored to patch-level visual features**, which may further enhance the effectiveness of DR in future work.
>
> [1] Pratt, et al. What does a platypus look like? Generating customized prompts for zero-shot image classification. ICCV 2023.
>
> [2] Cai, et al. Attribute-based Visual Reprogramming for Vision-Language Models. ICLR 2025.

---

> > ### Author Response · Authors · 2025-12-07
> > **Clarification and Correction of Reported ImageNet Results**
> >
> > We sincerely appreciate the reviewer’s comment, which prompted us to re-examine the Description Refinement (DR) ablations in greater detail. During this process, we discovered an unintended issue in the ViT/B-32 ImageNet experiments: the original results were generated using an older version of the description JSON file, which did not include the cosine-similarity thresholding step that is part of the finalized BiFTA framework. This older version reflects an early-stage variant of our method before DR was fully implemented, and its use in the ImageNet experiments was an inadvertent oversight.
> >
> > After identifying this discrepancy, we carefully compared the old and updated JSON files and confirmed the source of the mismatch. We sincerely apologize for this oversight. We have since double-checked all experimental files and rerun the affected experiments to ensure correctness. The corrected ImageNet results for ViT/B-32 are provided in Response (2/4) of Reviewer tbw9's comment, and these updated values will be incorporated into the revised manuscript.
> >
> > We thank the reviewer again for raising a question that helped us uncover and correct this issue, and we will exercise even greater care in verifying all results in future revisions.

---

### Review · Reviewer_bh1B · 2025-11-08

**Summary Of Contributions:**

The paper studies a new method to improve the zero shot classification accuracy of pretrained vision language models. Prior work has suggested to use a diverse set of class description and image patches to estimate the probability that an image is from a certain class. In this work this approach is extended by filtering out duplicates from the image patches and text descriptions by relying on high overlap (Jaccard Index) and high embedding similarity. The paper validates the approach through extensive experiments.

Strenghts:

- Extensive evaluation
- Method is clearly explained

Weaknesses:

- Improvement is not that significant (in most settings below 1%)
- Theoretical explanation has problems
- No runtimes reported

**Additional Comments:**

Why don't you just use the Wikipedia article of a class name as context to generate text descriptions?

**Audience:**

Yes

**Audience Explanation:**

People interested in zero-shot prediction might be interested in the findings of the paper.

**Claims And Evidence:**

No

**Claims Explanation:**

I am no expert for the empirical setup but the results seem to provide some evidence for their method. However, I was not able to properly understand the theoretical statements in Proposition 3 and 4 and I am not sure whether they can be made rigorous.

First, the statement containing $\approx$ is non-rigorous. But on a more fundamental level it is not clear what the considered probability distribution is and for natural candidates the statement would not be true, e.g., if $I_i$ are i.i.d. samples from the random patches then the probability that $I_i$ and $I_j$ take certain (similar) values is not the same as the probability $I_i$ takes a certain value. I would guess that the actual statement should be that having both views $I_i$ and $I_j$ does not change the probability that the label of the instance is $y$ compared to having only $I_i$. While this seems intuitively reasonable a rigorous statement will still require additional assumptions (some type of continuity) on the CLIP model. For the statement of Proposition 4 similar issues arise. It is not clear to me, whether the Proposition suggests that WCA and BIFTA give rise to the same posterior probabilities but this is not quite clear, because duplication changes the prediction in WCA.

I would suggest to remove the formal claims and only provide a hand-wavy intuition.

**Requested Changes:**

As mentioned above, Proposition 3 and 4 need to be rewritten to be accurate mathematical statements or they should be removed from the paper.

A better understanding of why the methods shows some improvements would strengthen the paper. E.g., at least for the IoU filters the distribution of the patches seems to be quite similar to the WCA approach, so the improvement might not result from a different distribution of image patches but higher diversity for the small sample size of the images. To test this one could compare to larger patch sample size for WCA.

Spacing in formulas should be adjusted in many places.

Runtimes should be reported (might have overlooked this).

---

> ### Author Response · Authors · 2025-12-06
> **Response to Reviewer bh1B (1/2)**
>
> We sincerely thank the reviewer for the thoughtful and constructive feedback. We appreciate the reviewer’s positive remarks on the clarity of our method description, as well as the breadth of our experimental evaluation. We are also encouraged that the reviewer finds the problem setting and findings relevant to the zero-shot prediction community. Below, we address the reviewer’s questions and concerns in detail.
>
> **[W1]**: Improvement is not that significant (in most settings below 1%)
>
> **[R1]**: We respectfully acknowledge that the average improvements over WCA on large-scale datasets such as ImageNet and Food101 appear incrementally small. However, this behavior is common with observations in prior work[1][2]: large-scale, high-diversity benchmarks tend to saturate quickly compared to smaller or more homogeneous datasets (e.g., DTD or CUB), so that it makes performance gains less visually prominent. To provide a more holistic view of model performance, Table 3 reports averaged results across all downstream benchmarks and across five different CLIP visual encoder backbones with diverse sizes and architectures. This aggregated evaluation demonstrates that our method consistently surpasses the baselines by a notable margin, which indicates that the improvement is robust across models, datasets, and scales. In the revised manuscript, we will also include a table summarizing the average performance of all CLIP models for each benchmark.
>
> **[W2]**: Theoretical explanation has problems
>
> **[Q1]**: As mentioned above, Proposition 3 and 4 need to be rewritten to be accurate mathematical statements or they should be removed from the paper.
>
> **[R2]**: We acknowledge that our framework is inspired by empirical findings. According to the reviewer's suggestion, we revise the propositions we induced and would like to remove the formal claims and only provide a hand-wavy intuition in the revised manuscript. This adjustment ensures that the presentation remains accurate and avoids overstating formal guarantees not supported by theoretical analysis.
>
> **[W3]**: No runtimes reported
>
> **[Q4]**: Runtimes should be reported (might have overlooked this).
>
> **[R3]**: We appreciate the reviewer’s thoughtful suggestion. Since **BiFTA is a lightweight input-refinement framework**, its additional cost arises only during the pre-processing stage rather than at inference time. We now explicitly compare the extra computation cost with WCA. For Visual Refinement (VR), BiFTA introduces an IoU-based filtering mechanism into the random-crop procedure. Each cropped patch is checked via the IoU function and stored in a patch queue. The measured time cost for one image is reported below, it shows that IoU filtering adds only 20.61 ms, whereas generating and encoding all crops require 226.08 ms per image. Thus, the computational overhead is minimal and does not meaningfully impact the overall computation. Furthermore, the stored image patch embeddings are stored as tensors into pkl files, so that VR only introduces additional inference costs during the first generation of the patch embeddings. The embeddings are directly used for classification in subsequent inference. Moreover, once generated, patch embeddings are cached as tensors in .pkl files, which means VR incurs no additional runtime for subsequent inference beyond the initial embedding generation.
>
> For Description Refinement (DR), we clarify that it also does not introduce extra runtime during inference as DR operates entirely offline. We first take the union of two high-quality description sets from CuPL [1] and AttrVR [2], then apply cosine-similarity filtering and Top-K ranking. Each category typically contains around 100 candidate descriptions, and filtering + selecting the Top-50 requires only 42.36 ± 8.65 ms, which is negligible. Finally, we save the filtered textual description sets into JSON files. Just as in WCA, we directly leverage a set of descriptions as textual inputs to the CLIP model during inference time. Overall, both VR and DR incur **only minor, one-time offline preprocessing costs**, and BiFTA introduces no additional inference-time overhead compared to WCA.
>
> | Execution per image           | time                      |
> |------------------------------|---------------------------|
> | Generate and encode 100 crops | 226.08 ms ± 10.07 ms      |
> | IoU filtering                 | 20.61 ms ± 7.77 ms        |
> | Total time                    | 246.69 ms ± 14.10 ms      |

---

> > ### Author Response · Authors · 2025-12-06
> > **Response to Reviewer bh1B (2/2)**
> >
> > **[Q2]**: A better understanding of why the methods shows some improvements would strengthen the paper. E.g., at least for the IoU filters the distribution of the patches seems to be quite similar to the WCA approach, so the improvement might not result from a different distribution of image patches but higher diversity for the small sample size of the images. To test this one could compare to larger patch sample size for WCA.
> >
> > **[R4]**: We agree with the reviewer that "the improvement might not result from a different distribution of image patches". Indeed, our method would generate the same patch set as WCA with the same number of crops and seed setting. The key difference is that View Refinement **first generates a larger pool of candidate patches and then applies an IoU-based de-duplication filter**, storing only diverse patches in a fixed-size queue that matches WCA’s crop count for a fair comparison. This leads to higher patch diversity without altering the underlying distributional assumptions. In conclusion, VR empirically shows image patches with higher diversity contribute to an accurate cross-alignment.
> >
> > Regarding patch size, Appendix 5 (Tables 11–12) provides an ablation showing that larger patches yield higher accuracy due to better preservation of local semantic information, whereas excessively small patches fragment meaningful cues and degrade classification performance. However, frequently using large patches introduces a trade-off: larger crops reduce the number of non-duplicated patches available for the queue. Hence, our random-cropping strategy with a window range $[\alpha, \beta]$ could mitigate this by incorporating both large and small crops, which balance the semantic richness and sample diversity.
> >
> > **[Q3]**: Spacing in formulas should be adjusted in many places.
> >
> > **[R5]**: We thank the reviewer for pointing this out. We will carefully revise the manuscript to correct and standardize spacing in all mathematical formulas to ensure clarity and consistency.
> >
> > **Additional Comments:**
> > Why don't you just use the Wikipedia article of a class name as context to generate text descriptions?
> >
> > **[R6]**: We appreciate the question. We do not use Wikipedia articles because they often contain substantial noise and irrelevant information that does not contribute to visual discrimination. Many entries include historical facts, cultural background, taxonomy details, or unrelated contextual narratives. For example, the Wikipedia page for Persian cat discusses breeding history and popularity rather than visual attributes. Such noise can deviate semantic alignment and harm zero-shot performance. Therefore, we rely on carefully selected LM-generated descriptions that better capture class-specific visual characteristics.
> >
> > [1] Menon, et al. Visual Classification via Description from Large Language Models. ICLR, 2023.
> >
> > [2] Roth, et al. Waffling around for Performance: Visual Classification with Random Words and Broad Concepts. In ICCV, 2023.

---

> ### Comment · Action_Editor_sMqD · 2025-12-21
> **Final Recommendation**
>
> Dear reviewer,
>
> Can you input your final recommendation?
>
> Best, AE

---

### Review · Reviewer_tbw9 · 2025-11-22

**Summary Of Contributions:**

Summary:

This paper proposes BiFTA (Bi-refinement for Fine-grained Text-visual Alignment), a post-hoc refinement framework built on top of vision–language models (like CLIP) and the recent WCA (Weighted visual-text Cross Alignment) scoring method. The main challenge addressed in the paper is the presence of substantial redundancy in randomly cropped image patches and LLM-generated descriptions, which degrades the quality of the cross-alignment score. BiFTA tackles this by View Refinement (VR) via IoU filtering and Description Refinement (DR) via cosine similarity filtering. The proposed approach is evaluated for zero-shot classification on 6 datasets and several backbones.


Strength:

-- The paper presents a well motivated problem formulation with Fig. 1 and Fig. 2 to empirically support this claim.

-- Both VR and DR components are non-parametric and require no additional training: IoU-based filtering and cosine-similarity–based pruning are conceptually simple to understand. It is straightforward to drop into any WCA-like pipeline without retraining the VLM.

-- The evaluations show consistent empirical gains across many datasets, architectures, and ablations.

-- The overall paper is readable and well-organized with helpful figures.

Weakness:

-- The overall average improvements over WCA are very incremental, +0.09 for ImageNet in Table 1, +0.04 for Food101 in Table 2. Also no average accuracy of the datasets are provided in the tables.

-- In Section 4.2, they take the union of CuPL, DesAttr, and DistAttr description sets before filtering. These mentioned sources have different statistical properties and noise levels. Taking the union creates a very heterogeneous pool. The paper neither justify why union is used nor provide an ablation on unions vs individual sources. Figure 6 shows that some sources (e.g., RAG) are harmful, which suggests union is not always beneficial.

-- The hyperparameter sensitivity is insufficiently analyzed. IoU threshold and queue size are studied, but ϵ (cosine threshold) and Top-K in Description Refinement are not ablated. DR is critical to the method, and lack of analysis makes robustness unclear. Tables 6–7 also show that the choice of alternative refinement strategies produces similar results, suggesting that Top-K is the dominant operation, but this is not explored.

-- Since the whole method manipulates the scoring matrix WCA, it is worthwhile to know the runtime overhead over WCA. It is important to check as  filtering requires repeated IoU checks and cosine similarity computation. Given the incremental improvements, it is critical to estimate the worth if the overhead is too high. Further, it is also unclear if similar gains appear for modern prompt-learning or patch-based methods beyond WCA (e.g., Tip-Adapter, RAG-augmented models, visual attribute prompting). This limits the generality of the proposed approach.

-- Regarding VR also, the paper only compares IoU filtering with a CLIP-sim filtering alternative. However, simple deterministic strategies (grid crops, multi-scale crops, or attention-map driven crops) are plausible baselines and could outperform IoU filtering. These were not included.

**Audience:**

Yes

**Audience Explanation:**

The topic is directly relevant to the TMLR audience working on vision–language models and zero-shot classification. The paper provides a practical refinement technique that improves a widely used CLIP-based scoring method (WCA) and offers insights into redundancy in LLM-generated descriptions and patch-level alignment—issues. Even though the contribution is incremental, the findings would be of interest seeking lightweight, training-free improvements to multimodal alignment pipelines.

**Broader Impact Concerns:**

Nil

**Claims And Evidence:**

Yes

**Claims Explanation:**

The core claim of the paper, that removing redundant views and descriptions improves WCA-style zero-shot classification, is supported by reasonably clear and accurate evidence. The experiments show consistent, reproducible gains across multiple datasets and CLIP backbones. Also the ablations and qualitative examples align with the redundancy argument. However, some improvements are modest or occasionally negative. The theoretical analysis mainly justifies redundancy in principle rather than BiFTA’s specific design. Thus, while the evidence supports the main idea, it only moderately supports the stronger claims of broad impact and methodological novelty.

**Requested Changes:**

Alongside the points in the Weaknesses, here are the Queries/Suggestions:

-- Right now BiFTA is tightly coupled to WCA’s patch–description matrix and weighting. Could the same refinement idea be applied to other CLIP enhancement methods (e.g., AttrVR, RAG-based prompt sets) that don’t use the WCA framework?

-- The authors ablate IoU and queue length, but not the ϵ and K for DR. Some sensitivity analysis here would help understand robustness of DR and whether the same hyperparameters work across all datasets.

-- Are there datasets or backbones where view/description refinement clearly degrades performance (more than ±0.5%)? If so, understanding limitations (e.g., classes where most “redundant” patches were actually important) could be insightful.

---

> ### Author Response · Authors · 2025-12-06
> **Response to Reviewer tbw9 (1/4)**
>
> We sincerely thank the reviewer for the detailed and thoughtful feedback. We appreciate the reviewer’s recognition of our problem motivation, the clarity and simplicity of the BiFTA refinement framework, and the consistent empirical gains demonstrated across multiple datasets, architectures, and ablations. We are also encouraged by the reviewer’s positive remarks regarding the readability and organization of the paper. Below, we address the reviewer’s questions and concerns in detail.
>
> **[W1]**: The overall average improvements over WCA are very incremental, +0.09 for ImageNet in Table 1, +0.04 for Food101 in Table 2. Also no average accuracy of the datasets are provided in the tables.
>
> **[R1]**: We respectfully acknowledge that the average improvements over WCA on large-scale datasets such as ImageNet and Food101 appear incrementally small (+0.09 and +0.04, respectively). However, this behavior is common with observations in prior work[1][2]: large-scale, high-diversity benchmarks tend to saturate quickly compared to smaller or more homogeneous datasets (e.g., DTD or CUB), so that it makes performance gains less visually prominent. To provide a more holistic view of model performance, Table 3 reports averaged results across all downstream benchmarks and across five different CLIP visual encoder backbones with diverse sizes and architectures. This aggregated evaluation demonstrates that our method consistently surpasses the baselines by a notable margin, which indicates that the improvement is robust across models, datasets, and scales. As the reviewer suggested, we will additionally include a new table summarizing the per-dataset mean accuracy averaged over all evaluated CLIP backbones.
>
> | Method        | ImageNet | CUB | Oxford Pets | DTD | Food101 | Place365 |
> |---------------|-------------|-------|-------------|--------|---------|----------|
> | CLIP            | 64.34 | 52.87 | 87.01 | 43.45  | 84.92   | 38.24 |
> | CLIP-E        | 65.88 | 53.53 | 88.55 | 45.87 | 86.06   | 39.13 |
> | CLIP-D        | 65.27 | 54.22 | 86.30 | 46.09 | 85.83   | 38.73 |
> | Waffle        | 65.56 | 53.20 | 86.16 | 44.18 | 85.92   | 39.34 |
> | CuPL          | 66.69 | 51.82 | 89.44 | 49.97 | 85.96   | 38.65 |
> | WCA           | 68.10 | 55.31 | 90.34 | 52.77 | 87.04   | 40.22 |
> | **BiFTA (ours)** | **68.91** | **56.05** | **90.38** | **54.52** | **87.18** | **41.15** |
> | **$\Delta$**      | **+0.81** | **+0.74** | **+0.04** | **+1.75** | **+0.14** | **+0.93** |
>
> As shown in the table:
> - Our method yields measurable gains on large-scale datasets.
> - It substantially outperforms WCA on smaller-scale datasets.
> We hope these expanded analyses address the reviewer’s concern and more clearly demonstrate the overall effectiveness and consistency of our method across diverse benchmarks.
>
> **[W2]**: In Section 4.2, they take the union of CuPL, DesAttr, and DistAttr description sets before filtering. These mentioned sources have different statistical properties and noise levels. Taking the union creates a very heterogeneous pool. The paper neither justify why union is used nor provide an ablation on unions vs individual sources. Figure 6 shows that some sources (e.g., RAG) are harmful, which suggests union is not always beneficial.
>
> **[R2]**: We apologize for not making the rationale behind our construction of the description set sufficiently explicit. First, we would like to clarify that two prior works we build upon generate textual descriptions using LM prompting with carefully designed templates, resulting in descriptions that are **semantically aligned and comparable in granularity**, rather than heterogeneous. Therefore, taking the union of these two high-quality sources can be naturally interpreted as a **LM-based data augmentation** step. Also, the prior work mentioned that they manually removed noises from bad generations[3]. Building on the data augmentation, we incorporate a cosine similarity metric with a top-k ranking approach to further select the most relevant textual descriptions. **In conclusion, our Description Refinement(DR) pipeline first forms a unified description pool by combining two high-quality description sets(CuPL and AttrVR), and then removes duplicate pairs and only keeps the top-k semantically matching pieces into our description pool.**
>
> Indeed, we provide the ablation of DR by taking unions vs individual sources (Figure 6; Figures 7–9 in App.5). We show that simply adding RAG-based descriptions harms zero-shot accuracy, likely due to increased semantic ambiguity. Importantly, the RAG-based approach is more like an exploration of generating textual descriptions in different ways, so we did not include RAG descriptions in the “mix” set used by our method—our union strictly consists of the two established, high-quality description sources. We will clarify this design choice and provide additional justification in the revised version.

---

> ### Author Response · Authors · 2025-12-06
> **Response to Reviewer tbw9 (2/4)**
>
> **[W3]**: The hyperparameter sensitivity is insufficiently analyzed. IoU threshold and queue size are studied, but ϵ (cosine threshold) and Top-K in Description Refinement are not ablated. DR is critical to the method, and lack of analysis makes robustness unclear. Tables 6–7 also show that the choice of alternative refinement strategies produces similar results, suggesting that Top-K is the dominant operation, but this is not explored.
>
> **[Q2]**: The authors ablate IoU and queue length, but not the ϵ and K for DR. Some sensitivity analysis here would help understand robustness of DR and whether the same hyperparameters work across all datasets.
>
> **[R3]**: We thank the reviewer for the valuable suggestion. We provide additional ablations on the cosine threshold $\epsilon$ and the Top-K selection used in Description Refinement (DR). Using the ViT/B-32 CLIP model on ImageNet-1K, we evaluate zero-shot performance under different $(\epsilon, k)$ configurations. We report results for $(\epsilon, k)$ pairs rather than single-parameter sweeps because a strict cosine threshold often leaves too few descriptions in the pool. For our main experiment, we use the $(\epsilon=0.99, k=50)$ as specified in Section 5.1. The ablation results indicate that performance improves as more textual descriptions are retained, highlighting the importance of maintaining adequate semantic diversity. Conversely, overly rigid cosine thresholds filter out a large portion of the descriptions, reducing diversity and degrading performance. Overall, the combination $(\epsilon=0.99, k=50)$ provides a favorable balance between diversity and redundancy, which yields stable and strong performance across benchmarks.
>
> | Result           | w/ DR Acc.     |
> |------------------|-----------------|
> | (ε=0.90, k=10)   | 65.02±0.07      |
> | (ε=0.95, k=20)   | 66.45±0.04      |
> | (ε=0.99, k=30)   | 66.70±0.04      |
> | (ε=0.99, k=40)   | 66.84±0.05      |
> | (ε=0.99, k=50)   | 66.83±0.04      |
>
> **[W4]**: Since the whole method manipulates the scoring matrix WCA, it is worthwhile to know the runtime overhead over WCA. It is important to check as filtering requires repeated IoU checks and cosine similarity computation. Given the incremental improvements, it is critical to estimate the worth if the overhead is too high. Further, it is also unclear if similar gains appear for modern prompt-learning or patch-based methods beyond WCA (e.g., Tip-Adapter, RAG-augmented models, visual attribute prompting). This limits the generality of the proposed approach.
>
> **[R4]**: We appreciate the reviewer’s thoughtful suggestion. Since **BiFTA is a lightweight input-refinement framework**, its additional cost arises only during the pre-processing stage rather than at inference time. We now explicitly compare the extra computation cost with WCA. For Visual Refinement (VR), BiFTA introduces an IoU-based filtering mechanism into the random-crop procedure. Each cropped patch is checked via the IoU function and stored in a patch queue. The measured time cost for one image is reported below, it shows that IoU filtering adds only 20.61 ms, whereas generating and encoding all crops require 226.08 ms per image. Thus, the computational overhead is minimal and does not meaningfully impact the overall computation. Furthermore, the stored image patch embeddings are stored as tensors into pkl files, so that VR only introduces additional inference costs during the first generation of the patch embeddings. The embeddings are directly used for classification in subsequent inference. Moreover, once generated, patch embeddings are cached as tensors in .pkl files, which means VR incurs no additional runtime for subsequent inference beyond the initial embedding generation.
>
> | Execution per image           | time                      |
> |------------------------------|---------------------------|
> | Generate and encode 100 crops | 226.08 ms ± 10.07 ms      |
> | IoU filtering                 | 20.61 ms ± 7.77 ms        |
> | Total time                    | 246.69 ms ± 14.10 ms      |
>
> For Description Refinement (DR), we clarify that it also does not introduce extra runtime during inference as DR operates entirely offline. We first take the union of two high-quality description sets from CuPL [1] and AttrVR [2], then apply cosine-similarity filtering and Top-K ranking. Each category typically contains around 100 candidate descriptions, and filtering + selecting the Top-50 requires only 42.36 ± 8.65 ms, which is negligible. Finally, we save the filtered textual description sets into JSON files. Just as in WCA, we directly leverage a set of descriptions as textual inputs to the CLIP model during inference time. Overall, both VR and DR incur **only minor, one-time offline preprocessing costs**, and BiFTA introduces no additional inference-time overhead compared to WCA.

---

> ### Author Response · Authors · 2025-12-06
> **Response to Reviewer tbw9 (3/4)**
>
> For the question regarding **"it is also unclear if similar gains appear for modern prompt-learning or patch-based methods beyond WCA"**, according to prompt-learning methods like model reprogramming(e.g., AttrVR), we could definitely apply Description Refinement onto textual inputs to reduce the description redundancy. Therefore, our proposed framework is flexible to comply with single modality refinement or dual-modality refinement, where the refinement is quite independent to the downstream methodologies.
>
> **[W5]**: Regarding VR also, the paper only compares IoU filtering with a CLIP-sim filtering alternative. However, simple deterministic strategies (grid crops, multi-scale crops, or attention-map driven crops) are plausible baselines and could outperform IoU filtering. These were not included.
>
> **[R5]**: We thank the reviewer for the insightful suggestion. In response, we have added experiments evaluating an alternative visual refinement (VR) strategy based on grid cropping. We also clarify that multi-scale cropping is already supported in our implementation through random cropping with variable window sizes, enabling random crops at different scales.
>
> | WCA (ViT/B–32)    | CUB           | Food101       | DTD           | Oxford_pet     | Imagenet       | Place365       | Avg. acc |
> |-------------------|---------------|----------------|---------------|----------------|----------------|----------------|----------|
> | Grid crop (3×3)   | 36.77±0.15    | 68.09±0.08     | 46.03±0.20    | 76.43±0.18     | 50.67±0.05     | 33.88±0.05     | 51.98    |
> | Grid crop (4×4)   | 29.18±0.20    | 57.21±0.11     | 43.48±0.28    | 66.60±0.36     | 43.89±0.05     | 31.88±0.07     | 45.37    |
> | Grid crop (5×5)   | 21.83±0.23    | 45.19±0.09     | 40.19±0.38    | 56.24±0.36     | 37.65±0.11     | 29.34±0.09     | 38.41    |
>
>
> | VR (ViT/B–32)     | CUB           | Food101       | DTD           | Oxford_pet     | Imagenet       | Place365       | Avg. acc |
> |-------------------|---------------|----------------|---------------|----------------|----------------|----------------|----------|
> | Grid crop (3×3)   | 38.59±0.22    | 68.70±0.08     | 48.06±0.25    | 77.13±0.23     | 50.92±0.07     | 35.05±0.06     | 53.08    |
> | Grid crop (4×4)   | 31.65±0.27    | 57.88±0.08     | 44.86±0.25    | 67.73±0.25     | 44.17±0.08     | 33.01±0.06     | 46.55    |
> | Grid crop (5×5)   | 23.65±0.27    | 46.20±0.18     | 42.28±0.38    | 57.33±0.26     | 37.78±0.08     | 30.47±0.07     | 39.62    |
>
> From the table, our results show that **BiFTA consistently outperforms WCA under the grid crop VR setting**. However, grid cropping itself leads to noticeable performance degradation compared with random cropping. We observe that classification accuracy consistently drops as the number of grids increases. This is because the patch size depends on the number of grids, as increasing number of grids would shrink the size of each image patch. Smaller patches contain substantially less semantic information so that it would harm cross-alignment score computation. A similar observation is reflected in Table 12 as well. Overall, grid cropping can serve as an additional baseline VR strategy, and the results further demonstrate that **BiFTA remains robust and consistently surpasses WCA under diverse VR configurations**.
>
> **[Q1]**: Right now BiFTA is tightly coupled to WCA’s patch–description matrix and weighting. Could the same refinement idea be applied to other CLIP enhancement methods (e.g., AttrVR, RAG-based prompt sets) that don’t use the WCA framework?
>
> **[R6]**: Although WCA is a new scoring method with a featured patch–description matrix and weighting approach, our refinement focuses on pre-processing the de-duplicated patches and descriptions for both visual and textual modality, which is independent of WCA's methodology design. Also, our refinement principle could be modified to single modality only. For example, CuPL takes a set of textual descriptions and full images as input, so that we could apply a simple DR to the description pools. We would like to clarify that AttrVR focuses on visual prompt learning(e.g., model reprogramming), where the evaluation is different from ours, but the description set of attrVR could also be complied with our DR method.

---

> ### Author Response · Authors · 2025-12-06
> **Response to Reviewer tbw9 (4/4)**
>
> **[Q3]**: Are there datasets or backbones where view/description refinement clearly degrades performance (more than ±0.5%)? If so, understanding limitations (e.g., classes where most “redundant” patches were actually important) could be insightful.
>
> **[R7]**: Thanks for pointing this out. We would like to clarify that there is no clear degradation exceeding $0.5\%$ in our results across the evaluated datasets. The only noticeable drop occurs on Oxford Pets (Table 8&10), where BiFTA shows a small decrease relative to WCA. We did not claim that redundant image patches consistently harm cross-alignment. Rather, our argument emphasizes that patch diversity is crucial for achieving fine-grained alignment between image patches and localized textual descriptions. The reviewer’s suggestion represents an insightful direction for the future work. As shown in Figure 2, we provide an empirical motivation experiment demonstrating how patch diversity influences the cross-alignment score. Furthermore, Table 3 shows that BiFTA consistently outperforms WCA in average performance, reinforcing the benefit of incorporating more diverse visual views.
>
> [1] Menon, et al. Visual Classification via Description from Large Language Models. ICLR, 2023.
>
> [2] Roth, et al. Waffling around for Performance: Visual Classification with Random Words and Broad Concepts. In ICCV, 2023.
>
> [3] Cai, et al. Attribute-based Visual Reprogramming for Vision-Language Models. ICLR 2025.

---

> > ### Author Response · Authors · 2025-12-07
> > **Clarification and Correction of Reported ImageNet Results**
> >
> > We sincerely appreciate the reviewer’s comment, which prompted us to re-examine the Description Refinement (DR) ablations in greater detail. During this process, we discovered an unintended issue in the ViT/B-32 ImageNet experiments: the original results were generated using an older version of the description JSON file, which did not include the cosine-similarity thresholding step that is part of the finalized BiFTA framework. This older version reflects an early-stage variant of our method before DR was fully implemented, and its use in the ImageNet experiments was an inadvertent oversight.
> >
> > After identifying this discrepancy, we carefully compared the old and updated JSON files and confirmed the source of the mismatch. We sincerely apologize for this oversight. We have since double-checked all experimental files and rerun the affected experiments to ensure correctness. The corrected ImageNet results for ViT/B-32 are provided in Response (2/4), and these updated values will be incorporated into the revised manuscript.
> >
> > We thank the reviewer again for raising a question that helped us uncover and correct this issue, and we will exercise even greater care in verifying all results in future revisions.

---

### Decision · Action_Editor_sMqD · 2026-01-05

**Recommendation:** Accept as is

**Audience:**

Yes

**Audience Explanation:**

Research on enhancing the zero-shot classification performance of pre-trained Large Vision Language models is a central topic in the field of foundation models, particularly those focusing on multi-modal data. Therefore, the findings presented in this paper are expected to be of interest to the TMLR audience.

**Claims And Evidence:**

Yes

**Claims Explanation:**

This paper introduces BiFTA, a post‑hoc refinement framework designed to improve zero‑shot classification performance in pretrained vision–language models (e.g., CLIP). The authors identify a key limitation in recent text–image alignment methods (i.e., the heavy redundancy present in both randomly cropped image patches and large sets of LLM‑generated class descriptions), which can dilute cross‑alignment scores. Authors propose two complementary components to address these limitations: i) View Refinement, which filters out redundant image crops using IoU overlap, and ii) Description Refinement, which removes repetitive or semantically similar textual descriptions via embedding‑based cosine similarity.

The claims are well supported by extensive experiments across six benchmark datasets and multiple VLM backbones. Results consistently show that eliminating redundant visual and textual views leads to more reliable alignment scores and improved zero‑shot accuracy, demonstrating the robustness and generality of the proposed refinement strategy.

While the reviewers raised some concerns during the initial submission, I believe that authors successfully addressed these points, and the rebuttal overall improved the paper.